# Nuclear receptors NHR-49 and NHR-79 promote peroxisome proliferation to compensate for aldehyde dehydrogenase deficiency in *C. elegans*

Lidan Zeng[1], Xuesong Li[2], Christopher B. Preusch[1], Gary J. He[1], Ningyi Xu[3], Tom H. Cheung[1,4,5], Jianan Qu[2], Ho Yi Mak[1]*

1 Division of Life Science, The Hong Kong University of Science and Technology, Hong Kong SAR, China, 2 Biophotonics Research Laboratory, Department of Electronic and Computer Engineering, The Hong Kong University of Science and Technology, Hong Kong SAR, China, 3 The MOE Key Laboratory of Biosystems Homeostasis & Protection and Innovation Center for Cell Signaling Network, Life Sciences Institute, Zhejiang University, Hangzhou, Zhejiang, China, 4 Center for Stem Cell Research, The Hong Kong University of Science and Technology, Hong Kong SAR, China, 5 State Key Laboratory in Molecular Neuroscience, The Hong Kong University of Science and Technology, Hong Kong SAR, China

* hym@ust.hk

**Data Availability Statement:** The data is publicly available and can be accessed via https://www.

## Abstract

The intracellular level of fatty aldehydes is tightly regulated by aldehyde dehydrogenases to minimize the formation of toxic lipid and protein adducts. Importantly, the dysregulation of aldehyde dehydrogenases has been implicated in neurologic disorder and cancer in humans. However, cellular responses to unresolved, elevated fatty aldehyde levels are poorly understood. Here, we report that ALH-4 is a *C. elegans* aldehyde dehydrogenase that specifically associates with the endoplasmic reticulum, mitochondria and peroxisomes. Based on lipidomic and imaging analysis, we show that the loss of ALH-4 increases fatty aldehyde levels and reduces fat storage. ALH-4 deficiency in the intestine, cell-nonautonomously induces NHR-49/NHR-79-dependent hypodermal peroxisome proliferation. This is accompanied by the upregulation of catalases and fatty acid catabolic enzymes, as indicated by RNA sequencing. Such a response is required to counteract ALH-4 deficiency since *alh-4; nhr-49* double mutant animals are sterile. Our work reveals unexpected inter-tissue communication of fatty aldehyde levels and suggests pharmacological modulation of peroxisome proliferation as a therapeutic strategy to tackle pathology related to excess fatty aldehydes.

## Author summary

Fatty aldehydes are generated during the turnover of membrane lipids and when cells are under oxidative stress. Because excess fatty aldehydes form toxic adducts with proteins and lipids, their levels are tightly controlled by a family of aldehyde dehydrogenases whose dysfunction has been implicated in genetic disease and cancer in humans. Here, we

ncbi.nlm.nih.gov/geo/query/acc.cgi?acc=
GSE162792.

**Funding:** The work was supported by Research
Grants Council GRF 16102118 to HYM. The funder
had no role in study design, data collection and
analysis, decision to publish, or preparation of the
manuscript.

**Competing interests:** The authors have declared
that no competing interests exist.

characterize mutant *C. elegans* that lack a conserved, membrane-associated aldehyde
dehydrogenase ALH-4. Despite elevated levels of fatty aldehydes, these mutant worms sur-
vive by increasing the abundance of peroxisomes, which are important organelles for lipid
metabolism. Such peroxisome proliferative response depends on the activation of tran-
scription factors NHR-49 and NHR-79, via putative endocrine signals. Accordingly, the
fertility of *alh-4* mutant worms relies on NHR-49. Our work suggests a latent mechanism
that may be activated during aldehyde dehydrogenase deficiency.

## Introduction

Glycerophospholipids and sphingolipids are major components of biological membranes [1].
The regulated turnover of their head groups and acyl chains alters membrane structural prop-
erties such as fluidity, thickness and curvature. In addition, the modification and cleavage of
acyl chains allow them to serve as precursors of signaling molecules. For example, membrane
derived polyunsaturated fatty acids are essential precursors of eicosanoids and endocannabi-
noids that modulate inflammatory response [2]. However, polyunsaturated acyl chains are also
prone to attack by reactive oxygen species (ROS), which are generated by metabolic activities
and environmental factors [3]. Such peroxidation of PUFAs generates short chain and
medium chain fatty aldehydes [4]. In addition, long chain fatty aldehydes can be derived from
the metabolism of sphingolipids, etherlipids and fatty alcohols [5]. Reactive fatty aldehydes
form covalent adducts with proteins, lipids and nucleic acids and contribute to oxidative stress
[6,7]. Therefore, the level of cellular fatty aldehydes is under tight regulation via the conserved
aldehyde dehydrogenase (ALDH) superfamily [8].

ALDHs limit cellular stress by catalyzing the NAD(P)+ dependent conversion of fatty alde-
hydes to fatty acids, which can then be fed into anabolic or catabolic pathways [9]. Unique
among ALDH family members is FALDH, also called ALDH3A2, because it associates with
the ER or peroxisomal membranes via a transmembrane helix [10,11]. ALDH3A2 catalyzes
the oxidation of fatty aldehydes that range from 6–24 carbons [12]. Besides ROS-induced alde-
hydes, FALDH processes fatty aldehydes from the metabolism of etherlipids (such as plasmalo-
gens), sphingolipids and dietary branched-chained lipids [8]. For example, it is required for
converting the sphingosine 1-phosphate metabolite hexadecenal to hexadecenoic acid in the
conversion of sphingolipid to glycerolipid [13].

Nuclear receptors are conserved metazoan transcription factors that are distinguished by
their C2H2 zinc finger-containing DNA binding domain and ligand binding domain [14]. In
humans, nuclear receptors, such as estrogen receptors, are known to bind steroid hormones
with high affinity. This is because of the tight fit between the ligand binding pocket with the
cognate ligand. However, there were also a number of nuclear receptors, called orphan recep-
tors, which had no known ligands when they were initially identified [15]. Subsequent efforts
led to the discovery of fatty acid or cholesterol metabolites as previously unrecognized ligands
to some of them. Prime examples include the peroxisome proliferator activated receptors
(PPARs). The structures of the ligand binding domain of PPARs reveal a large ligand binding
pocket that accommodates a range of fatty acid derivatives and synthetic ligands [16]. PPARs
form heterodimers with retinoid X receptors (RXRs) and bind to PPAR response elements
(PPREs) to modulate the expression of their target genes [17,18]. Based on genetic and phar-
macological studies in vitro, in mice and humans, PPARα and PPARγ have well-established
roles for energy homeostasis [19,20]. For example, PPARα is pivotal for the activation of genes
that are required for fatty acid uptake and turnover [21].

The nuclear receptor family is greatly expanded in *C. elegans*. In comparison to 48 nuclear receptors in humans, the *C. elegans* genome is predicted to encode 284 nuclear receptors [22]. To date, most of them remain uncharacterized and have no known ligands. Nevertheless, a handful of *C. elegans* nuclear receptors have been shown to regulate developmental dauer arrest, molting, aging or metabolism [23–30].

One of the well characterized *C. elegans* nuclear receptors is NHR-49. Although it bears relatively high sequence similarity to mammalian Hepatocyte Nuclear Factor 4 (HNF4), NHR-49 appears to serve similar functions as PPARα [31]. Notably, NHR-49 modulates gene expression programs that govern fatty acid catabolism, desaturation and remodeling under fed and fasted conditions [31–34]. As a result, NHR-49 has been implicated in cold adaptation of phospholipid membrane composition and preservation of germline stem cells in starved animals [35,36]. NHR-49 has also been shown to regulate lifespan in response to diffusible molecules [37,38]. NHR-49 partners with different nuclear receptors to regulate distinct sets of target genes [32]. Here, we report that NHR-49 acts with a previously uncharacterized partner, NHR-79 to promote peroxisome proliferation in mutant worms that lack ALH-4, which is the predicted ortholog of mammalian ALDH3A2/FALDH. Based on lipidomic analysis, ALH-4 deficient worms have elevated levels of fatty aldehydes and reduced levels of neutral lipids, phospholipids and sphingolipids. Interestingly, the reproduction of ALH-4 deficient worms was dependent on NHR-49 because *alh-4; nhr-49* mutant worms were sterile. Our results reveal a nuclear receptor coordinated peroxisome proliferative response, which effectively counters the accumulation of toxic fatty aldehydes.

## Results

### Loss of *alh-4* function suppresses lipid droplet expansion

The attenuation of peroxisomal β-oxidation causes excess accumulation of triacylglycerol (TAG) in expanded lipid droplets (LDs) in mutant animals that lack the DAF-22 thiolase [39]. This phenotype is most pronounced in the *C. elegans* intestine, which serves as a major fat storage tissue, and could be readily detected using quantitative Stimulated Raman Scattering (SRS) microscopy [40] (Fig 1A). To identify genetic regulators of fat storage and LD expansion, we conducted a forward genetic screen for suppressor mutants that reversed the *daf-22* mutant phenotypes. A complementation group with three recessive alleles (*hj27*, *hj28*, *hj29*) were identified. Following genetic mapping, targeted sequencing of *alh-4* revealed that *hj27* is a missense allele and *hj28* and *hj29* are nonsense alleles of *alh-4* (S1A Fig), which encodes a predicted aldehyde dehydrogenase that catalyzes fatty aldehyde into fatty acid. Using CRISPR-mediated genome editing [41], we generated an additional *alh-4* deletion allele, *hj221* (S1A Fig). We found that *alh-4(hj221)*, as well as RNA interference (RNAi) against *alh-4*, suppressed LD expansion of *daf-22(-)* mutant animals (S1B Fig). Therefore, *alh-4(hj29)* is likely to be a strong loss of function allele.

The loss of *alh-4* function increased the brood size and the rate of larval development of *daf-22(-)* worms (S1C–S1D Fig). Using SRS microscopy, we quantitatively measured the effect of ALH-4 deficiency on cellular fat storage. The loss of *alh-4* function suppressed fat accumulation and LD expansion of *daf-22(-)* worms (Fig 1A–1D). In addition, the *alh-4(hj29)* worms accumulated less fat than wild type worms (Fig 1A–1B). Accordingly, *alh-4(hj221)* (denoted as *alh-4(-)* hereafter) worms had significantly less TAG than wild type worms, based on liquid chromatography-mass spectrometry (LC-MS) analysis (Fig 1F and S3 Dataset). We observed no significant difference in the pharyngeal pumping rate of wild type and *alh-4* mutant worms (S1E Fig), suggesting that the reduction in fat storage was unlikely due to a change in food intake. To compare the size of intestinal LDs in wild type and mutant worms, we used an

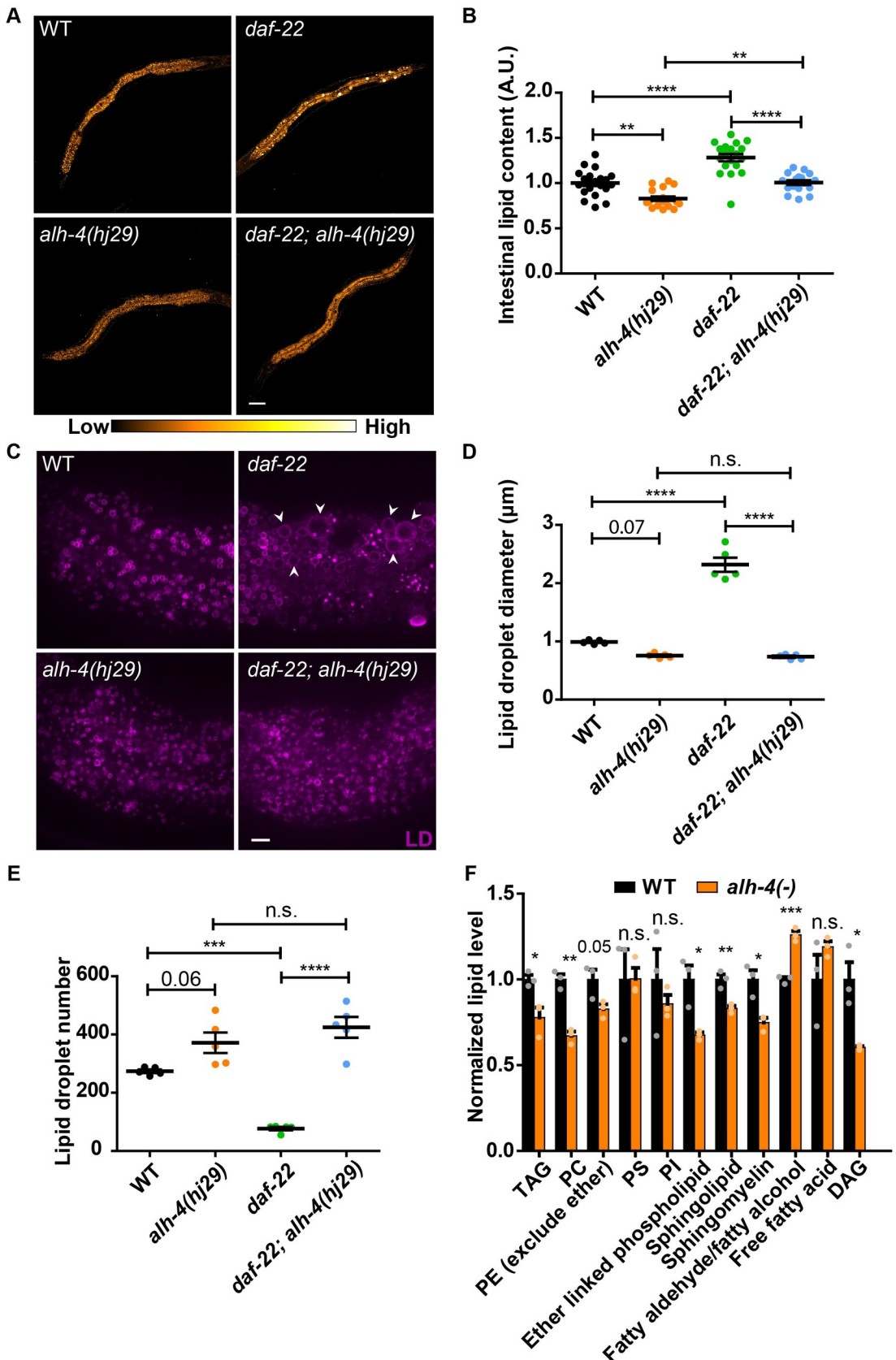

**Fig 1. ALH-4 regulates lipid storage and lipid profile.** (A) Representative Stimulated Raman Scattering (SRS) images of L4 worms of wild type (WT), *daf-22(ok693)*, *alh-4(hj29)* and *alh-4(hj29); daf-22(ok693)*. Scale bar = 50μm. (B) Quantification of average SRS signal intensity. n > 16 for all groups. (C) Representative images of lipid droplets in the second intestinal segment of L4 worms. Lipid droplets (LDs) are labeled with mRuby::DGAT-2*(hjSi112)*. Enlarged LDs are marked by arrowheads. Scale bar = 5μm. (D) Quantification of lipid droplet diameter in the second intestinal segment. n = 5 for all groups. (E) Quantification of total lipid droplet number in the second intestinal segment in individual worms. n = 5 for all groups. Two-way ANOVA with Tukey's multiple comparisons test was applied in B, D & E. (F) Lipidomic analysis of WT and *alh-4(-)* worms. Quantification of lipid species detected by LC-MS. The value of each species in wild type samples was set as 1. Each dot represents the average value of two technical replicates of each biological sample. n = 3 for all groups. TAG, triacylglycerol; PE, phosphatidylethanolamine; PC, phosphatidylcholine; PS, phosphatidylserine; PI, phosphoinositide; DAG, diacylglycerol. Two-tailed unpaired Student's t-test was applied in F. For all statistical tests, $^*p < 0.05$, $^{**}p < 0.01$, $^{***}p < 0.001$, $^{****}p < 0.0001$, the actual p-value is displayed when p is between 0.05 and 0.1, n.s. p > 0.1. In all plots, mean ± SEM of each group is shown.

established LD marker, mRuby::DGAT-2 [42]. The LD size profiles of *alh-4(hj29)* and *daf-22* (-); *alh-4(hj29)* worms almost completely overlapped (S1F Fig). The reduction of LD size was accompanied by an increase in LD number in *alh-4(hj29)* and *daf-22*(-); *alh-4(hj29)* worms, in comparison with wild type and *daf-22*(-) worms, respectively (Fig 1E). We conclude that the loss of *alh-4* function suppresses LD expansion without compromising LD biogenesis.

## Tissue specific subcellular localization of ALH-4

Based on sequence homology, ALH-4 appears to be closely related to human Class III aldehyde dehydrogenases (S3A Fig). In agreement with such prediction, LC-MS based lipidomic analysis indicated that *alh-4(-)* worms accumulated significantly more fatty aldehydes and its related metabolite fatty alcohols, than wild type worms (Figs 1F and S1G, and S3 Dataset). Furthermore, ALH-4 and human ALDH3A2 (also known as FALDH) are the only proteins in this class of enzymes that bear C-terminal tail anchors in the form of a single transmembrane alpha helix. Prior studies indicated that ALDH3A2 associates with peroxisomal membrane and the endoplasmic reticulum via distinct C-terminal tails that are encoded by alternatively spliced exons [10]. Similarly, we noted that three *alh-4* isoforms were annotated, based on publicly available RNA sequencing data (Fig 2A). These isoforms differ by their 3' ends, which encode alternative C-termini that follow a common transmembrane helix. As a first step of determining the expression pattern and intracellular localization of ALH-4 isoforms, we inserted the coding sequence of the green fluorescent protein (GFP) immediately after the start codon of *alh-4* by CRISPR-mediated genome editing (Fig 2A). As a result, all three isoforms of ALH-4 were expressed as GFP fusion proteins from the endogenous locus. This GFP knockin strain, *alh-4(hj179)*, retained normal ALH-4 function because LD expansion was unperturbed when *daf-22* was mutated, unlike what we observed in *daf-22; alh-4(-)* worms. We detected strong expression of functional GFP::ALH-4 in the intestine and hypodermis (Fig 2B–2F). Our results suggest that ALH-4 may be important in regulating lipid metabolism in these tissues.

We used a suite of red fluorescent organelle markers to determine the subcellular localization of GFP::ALH-4. Specifically, mRuby::DGAT-2, ACS-22::tagRFP, and TOMM-20(N):: mRuby, tagRFP::peroxisomal targeting signal 1(PTS1) were used to label LDs, ER, mitochondria and peroxisomes, respectively [43,44]. Live transgenic worms were observed using confocal microscopy. In the hypodermis, we detected co-localization of GFP::ALH-4 signals with red fluorescent signals that labeled mitochondria and peroxisomes (Fig 2B and 2C). In the intestine, GFP::ALH-4 signals co-localized with the ER marker and to a lesser extent, LD and mitochondrial markers (Fig 2D–2F). Therefore, ALH-4 appears to use tail anchors (S2 Fig and S1 Text) to associate with an overlapping set of organelles, where lipid anabolism, storage or catabolism take place. It is conceivable that the fatty acid products of ALH-4 can be channeled into metabolic pathways locally at respective organelles.

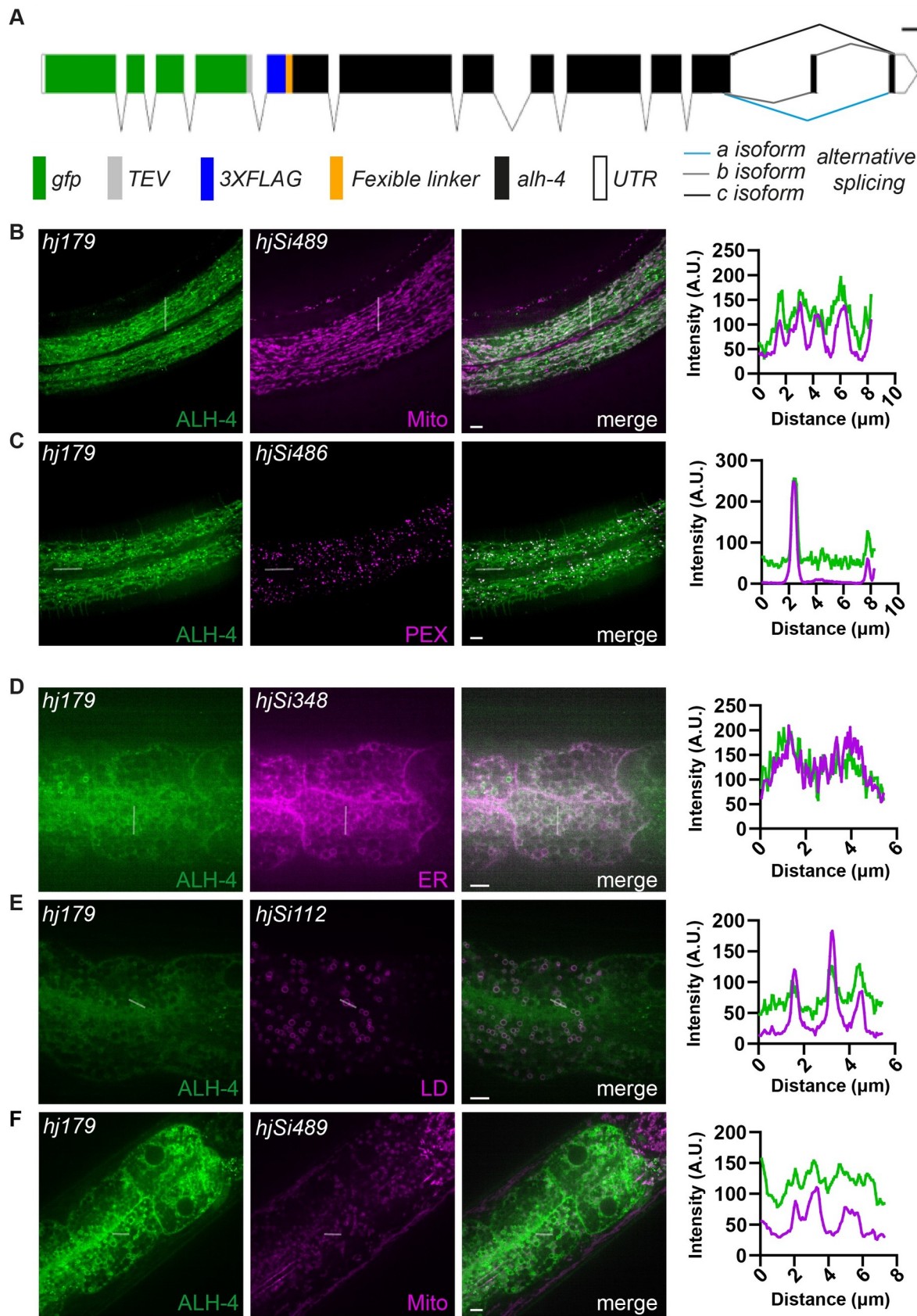

**Fig 2. Tissue-specific subcellular localization of ALH-4.** (A) Schematic representation of *gfp::alh-4(hj179)*. Scale bar = 100bp. Alternative splicing results in three ALH-4 isoforms that differ only at the C-terminus. (B-C) Representative images showing the subcellular localization of GFP::ALH-4 in the hypodermis (left). Fluorescence intensity profile from the indicated line scans (right). (B) GFP::ALH-4 largely colocalized with the mitochondria marker, TOMM-20N::mRuby (*hjSi489*). (C) The bight puncta of GFP::ALH-4 colocalized with the peroxisome marker, tagRFP::PTS1 (*hjSi486*). (D-F) Representative images showing the subcellular localization of GFP::ALH-4 in the intestine (left). Fluorescence intensity profile from the indicated line scans (right). (D) GFP::ALH-4 largely colocalized with the ER marker, ACS-22::tagRFP *(hjSi348)*. (E) Partial colocalization of GFP::ALH-4 with the lipid droplet marker mRuby::DGAT-2 (*hjSi112*). (F) GFP:: ALH-4 did not colocalize with the mitochondria marker, TOMM-20N::mRuby (*hjSi489*). Scale bar = 5μm. Mito, mitochondria; PEX, peroxisome; ER, endoplasmic reticulum; LD, lipid droplet.

## The transmembrane domain and the gate-keeper helix are required for ALH-4 function

Having established that ALH-4 associates with organelle membranes with the same mechanism as ALDH3A2, we next sought to determine if ALH-4 and ALDH3A2 share orthologous function. To this end, we expressed a chimeric protein that harbors the ALDH3A2 cytosolic catalytic domain (a.a. 1–465) and the predicted transmembrane helix and C-termini of ALH-4 in *alh-4(-)* worms (S3B Fig). Our strategy ensured that the ALDH3A2 catalytic domain was targeted to appropriate organelles in *C. elegans*. Intestinal expression of the chimeric protein, from two independent transgenes, cell-autonomously restored the LD size of *alh-4(-)* worms (Fig 3A–3C). Our results suggest that ALH-4 and ALDH3A2 most likely share the same function in converting fatty aldehydes to fatty acids, possibly with similar substrate specificity.

Its broad association with membranous organelles led us to ask if membrane association was required for ALH-4 function. Starting with the GFP knockin strain, *alh-4(hj179)*, which expressed full-length GFP::ALH-4 fusion protein from the endogenous locus, we used CRISPR to delete the sequence encoding the transmembrane helix and C-termini of the *alh-4* gene (S3B–S3C Fig). The GFP::ALH-4(ΔC) truncated protein retained the entire catalytic domain of ALH-4. As expected, the GFP fusion protein was cytosolic in both intestine and hypodermis (Fig 3D). The expression level of GFP::ALH-4(ΔC) was comparable to GFP::ALH-4 based on the intensity of GFP fluorescence. To assess the function of GFP::ALH-4(ΔC), we used mRuby::DGAT-2 as the LD marker. The average LD size was significantly reduced in animals that expressed GFP::ALH-4(ΔC), in comparison to those that expressed wild type GFP::ALH-4 (Fig 3E–3F). The mutant phenotype was not as pronounced as in *alh-4(hj29)* worms, suggesting that the function of GFP::ALH-4(ΔC) was severely compromised but not eliminated.

Structure-function analysis of ALDH3A2 indicated that a 'gate-keeper' helix is responsible for its substrate selectivity toward long-chain fatty aldehydes in vitro [45]. Deletion of the gate-keeper helix in purified ALDH3A2 strongly reduced its catalytic capacity to long-chain dodecanal and hexadecanal, but not medium-chain octanal [45]. The helix is located immediately N-terminal to the transmembrane helix, and appears to be conserved in ALH-4 (S3B Fig). Therefore, we tested if the gate-keeper helix is required for ALH-4 function in vivo, by deleting its coding sequence from the GFP knockin strain, *alh-4(hj179)*. We found that the organelle targeting of GFP::ALH-4(ΔGK) was unperturbed in the intestine and hypodermis (Fig 3D). In addition, the expression level of GFP::ALH-4(ΔGK) was comparable to GFP::ALH-4 based on the intensity of GFP fluorescence. However, worms expressing GFP::ALH-4(ΔGK) had significantly smaller LDs when compared with those expressing GFP::ALH-4 (Fig 3E and 3F). Our results show that the C-terminal transmembrane domain and the gate-keeper helix are both required for ALH-4 function in vivo. It should be noted that our deletion mutants were designed based on the ALDH3A2 structure, in order to prevent the disruption of globular domains. However, we cannot rule out folding defects of these mutants, which might contribute to their functional deficit. Additional structural studies will be required to address this issue further.

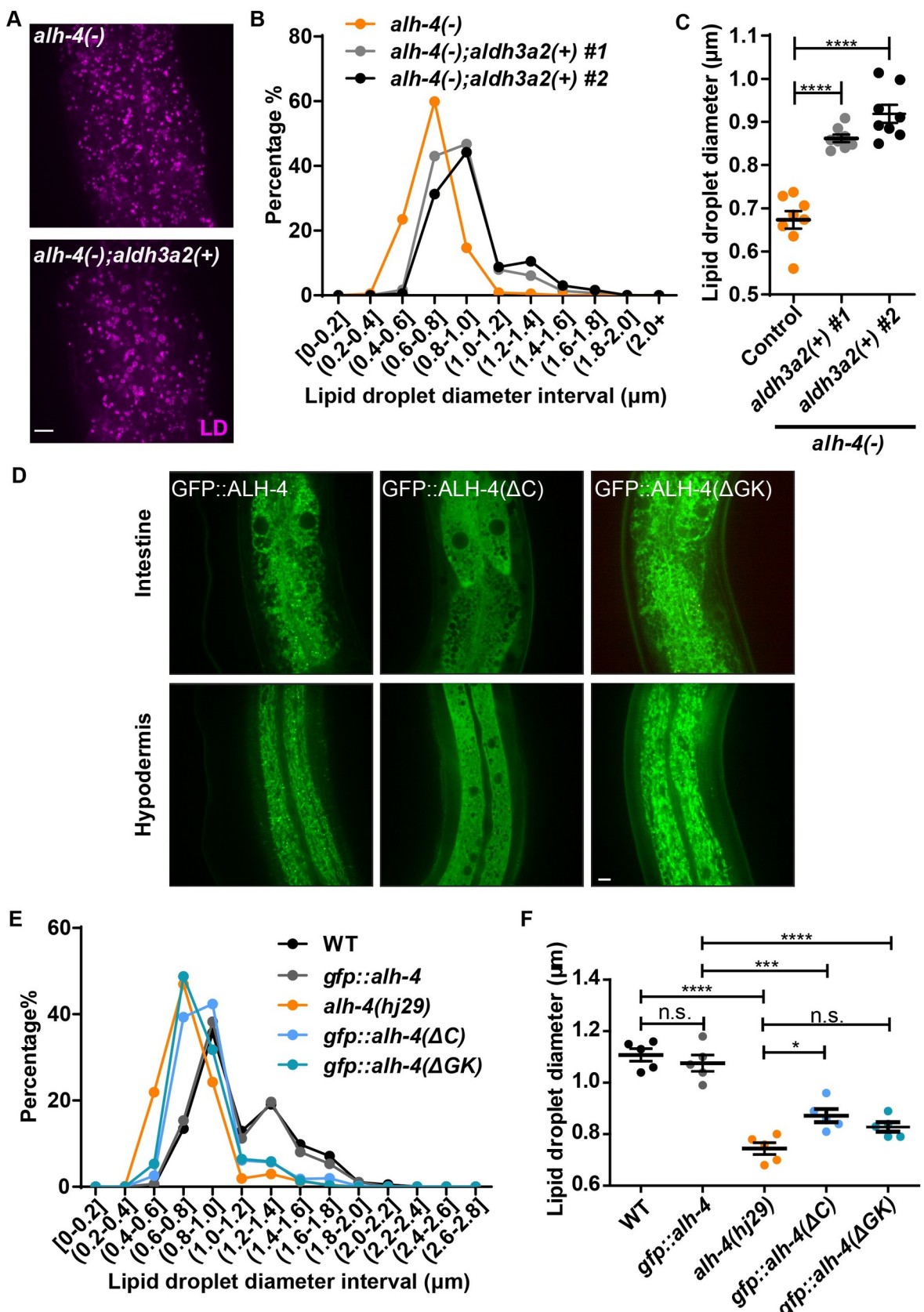

**Fig 3. Similar to human ALDH3A2, the membrane anchor and gatekeeper regions are necessary for proper ALH-4 function. (A-C)** Lipid droplets in the second intestinal segment of *alh-4(-)* L4 worms with or without independent extrachromosomal arrays expressing chimeric human ALDH3A2 in the intestine (*hjEx27, hjEx28 [vha-6p::GFP::aldh3a2(1-463aa)::alh-4(464aa-stop)]*). (A) Representative images of lipid droplets labeled with DHS-3::mRuby (*hj200*). Scale bar = 5μm. (B) The size distribution of lipid droplets in the second intestinal segment. (C) Quantification of average lipid droplet diameter. n = 8 for all groups. Two independent lines were quantified. (D) Representative images showing the subcellular localization of GFP tagged wild type ALH-4 (*hj179*) (left), cytosolic ALH-4 harboring a C-terminal deletion (ALH-4ΔC, *hj261*) (middle) and ALH-4 with a deletion of the gatekeeper region (ALH-4ΔGK, *hj264*) (right). Organelle-specific targeting was not affected by the deletion of the gatekeeper region. Scale bar = 5μm. (E) The size distribution of lipid droplets in the second intestinal segment. mRuby::DGAT-2 (*hjSi112*) was used to label lipid droplets. (F) Quantification of lipid droplet size in the second intestinal segment. For all statistical tests, *p < 0.05, **p < 0.01, ***p < 0.001, ****p < 0.0001, the actual p-value is displayed when p is between 0.05 and 0.1, n.s. (not significant) p > 0.1 (one-way ANOVA with Tukey's multiple comparisons test). In all plots, mean ± SEM of each group is shown.

## Upregulation of a peroxisome-related gene expression program in ALH-4 deficient worms

The membrane anchored aldehyde dehydrogenase family, to which ALH-4 belongs, is known to play a critical role in converting membrane associated long-chain aliphatic aldehydes to fatty acids [46]. Such conversion is critical to resolve oxidative stress-induced lipid peroxidation and turnover reactive lipids for their utilization in membrane composition, energy production, or storage. Therefore, we hypothesized that ALH-4 deficient worms might be subject to elevated oxidative stress and became susceptible to exogenous stress inducers. To this end, we compared wild type and *alh-4(-)* worms for their sensitivity to hydrogen peroxide and paraquat. We found that *alh-4(-)* worms had comparable sensitivity to paraquat as wild type worms (S4A Fig). Surprisingly, *alh-4(-)* worms were more resistant to hydrogen peroxide than wild type worms, and showed enhanced survival after being exposed to 0.75mM hydrogen peroxide for 4 hours (S4B Fig). Our results suggest that *alh-4(-)* worms might mount an anti-oxidative response to counter the accumulation of endogenous reactive lipids. Such a hormetic response renders these worms more resistant to some but not all exogenous stress inducers. To elucidate the molecular basis of such response, we compared the gene expression profiles of wild type and *alh-4(-)* worms by high throughput RNA sequencing.

We classified statistically significant, differentially expressed genes, using the g:Profiler analysis platform [47]. We found that the differentially expressed genes were highly enriched under the KEGG category of 'peroxisome' (Fig 4A, and S1 and S2 Datasets). This was followed by categories that were broadly related to fatty acid metabolism. Accordingly, a suite of genes that encode enzymes for peroxisomal function, including fatty acid β-oxidation, were significantly upregulated in *alh-4(-)* worms (Fig 4B and 4C). We also noted the concerted induction of *ctl-1*, *ctl-2*, and *ctl-3*, which encode peroxisomal catalases that are known to neutralize hydrogen peroxide, generated during peroxisomal fatty acid β-oxidation (Fig 4C).

Next, we sought to correlate the upregulation of peroxisome related genes in *alh-4(-)* worms with an increase in peroxisome number or function. We focused on the intestine and hypodermis, where *alh-4* was primarily expressed. To this end, we generated tissue-specific, single-copy transgenic reporter strains, which expressed red fluorescent tagRFP that was targeted to the peroxisomal matrix by the peroxisomal targeting signal PTS1. We found that the density of peroxisomes was significantly increased in the hypodermis, but not the intestine (Figs 4D and S4C). To extend this observation, we chose *daf-22* as another endogenous reporter gene, because its expression was induced in *alh-4(-)* worms according to our RNA sequencing data (Fig 4C). We inserted the coding sequence of tagRFP into *daf-22* by CRISPR, to yield transgenic worms that expressed a functional tagRFP::DAF-22 fusion protein from its endogenous locus. We observed an overall 2-fold increase of tagRFP::DAF-22 fluorescence signal in *alh-4(-)* worms (Fig 4E). Taken together, we propose that the enhanced resistance of

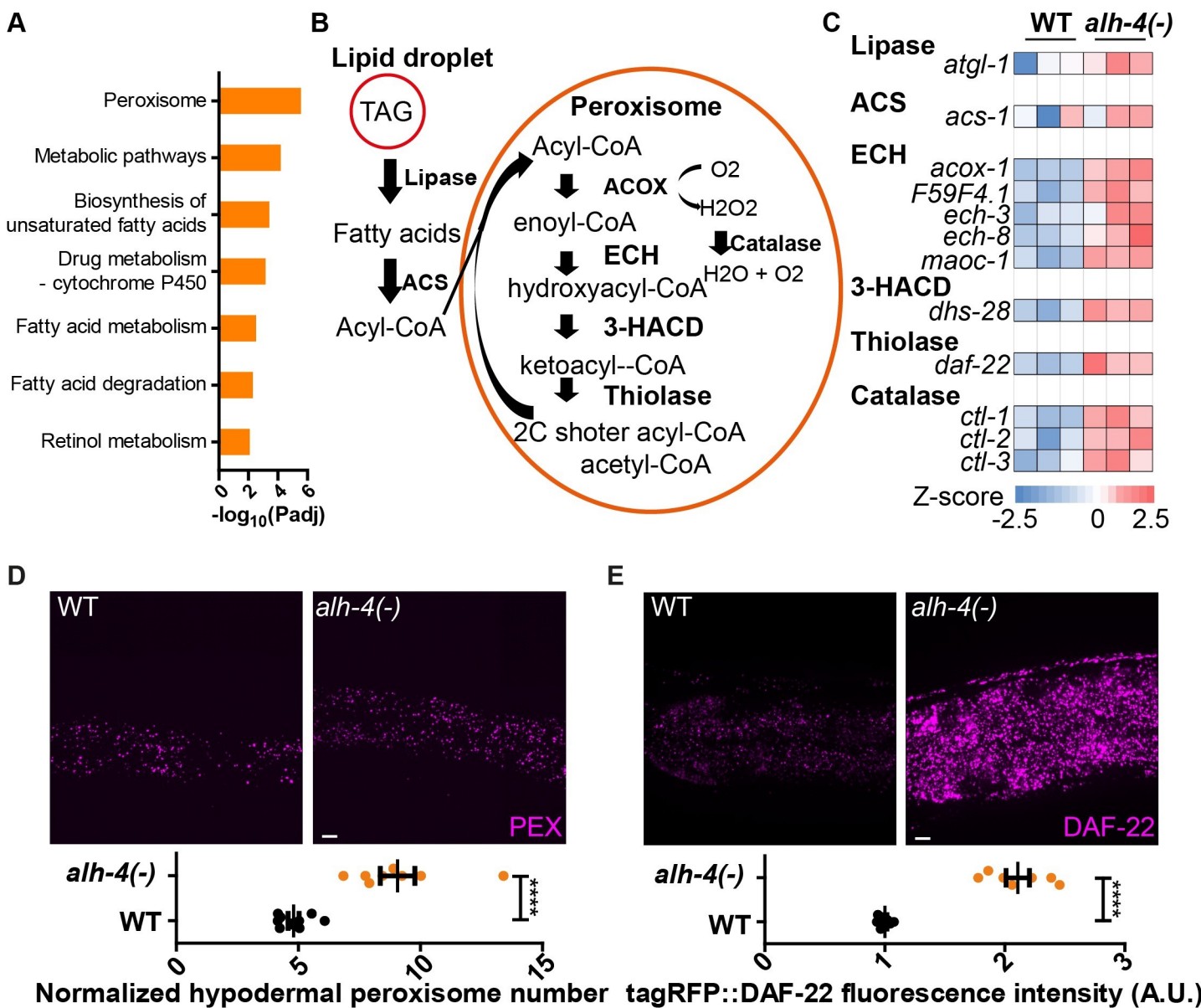

**Fig 4. Induction of peroxisomal fatty acid catabolism in *alh-4* mutants.** (A) KEGG pathway analysis of the differentially expressed genes (fold change > 2 and p-value < 0.05, negative binomial test) between wild type (WT) and *alh-4(-)* worms. The analysis was performed with https://biit.cs.ut.ee/gprofiler/gost. The peroxisome pathway and additional pathways related to lipid metabolism were returned as top hits. (B) Schematic representation of the peroxisomal fatty acid β-oxidation pathway. At the top left corner, triacylglycerols (TAG) in the lipid droplet are hydrolyzed into free fatty acids by lipases. The fatty acids are converted to fatty acyl-CoA by acyl-CoA synthetases. Acyl-CoA oxidases pass electron directly to oxygen and produce hydrogen peroxide ($H_2O_2$) while introducing the double bond to fatty acyl-CoA. Hydrogen peroxide is converted to oxygen and water via catalases. ACS, acyl-CoA synthetase; ACOX, acyl-CoA oxidase; ECH, enoyl-CoA hydratase; HACD, hydroxyalkyl-CoA dehydrogenase. (C) Heatmap showing the differential expression of genes encoding peroxisomal fatty acid β-oxidation enzymes in WT and *alh-4* worms. The FPKM values are converted to Z-scores. High expression is indicated with red, while low expression is indicated with blue. n = 3 for each group. Only genes showing significant difference (p-value < 0.1) in the expression are shown except *acs-1* (p-value = 0.30). (D) Representative images of hypodermal peroxisomes labeled by tagRFP::PTS1 (*hjSi486*) in WT and *alh-4(-)* worms (top). Quantification of hypodermal peroxisome number (bottom). (E) Representative images showing the fluorescence signal of DAF-22 tagged with tagRFP (*hj234*) in WT and *alh-4(-)* worms (top). Quantification of fluorescence signals (bottom). PEX, peroxisome; LD, lipid droplet. Scale bar = 5μm. In all plots, mean ± SEM of each group is shown. For all statistical test, *p < 0.05, **p < 0.01, ***p < 0.001, ****p < 0.0001, the actual p-value is displayed when p is between 0.05 and 0.1, n.s. (not significant) p > 0.1 (two-tailed unpaired Student's t-test).

*alh-4(-)* worms to hydrogen peroxide is collateral to peroxisome proliferation and the elevation of peroxisomal activity.

## Peroxisome proliferation is independent of ATGL-1 directed lipolysis in *alh-4* mutant worms

From our gene expression analysis, we noted that *atgl-1*, which encodes the *C. elegans* adipose triglyceride lipase (ATGL) ortholog, was slightly induced in *alh-4(-)* worms (Fig 4B and 4C). ATGL hydrolyzes a fatty acyl chain from TAG to release fatty acids [48]. Fatty acids can then be activated by acyl-CoA synthetases (ACS) to form acyl-CoA before being channeled to anabolic or catabolic pathways, such as β-oxidation. Besides, lipolytic products as well as exogenous fatty acids have been reported to activate PPARs (Peroxisome Proliferator-Activated Receptors) to induce the expression of genes for fatty acid oxidation [49]. It is plausible that fatty acids released from LDs by ATGL-1 can serve as signaling molecules that modulate peroxisome proliferation and function in *alh-4(-)* worms. To test this hypothesis, we knocked down *atgl-1* by RNAi in both WT and *alh-4(-)* worms. We observed an increase in average intestinal LD size, to a similar extent, in both cases (S4D and S4E Fig). Therefore, ATGL-1 appeared to work in parallel of ALH-4. Furthermore, we found that RNAi against *atgl-1* did not alter *daf-22* expression or the abundance of hypodermal peroxisomes (S4F–S4G Fig). Our results indicate that lipolysis is not required for the transcriptional response toward ALH-4 deficiency.

## NHR-49 and NHR-79 are required for the transcriptional response to ALH-4 deficiency

To identify transcription factors that are responsible for the altered gene expression program in *alh-4(-)* worms, we focused our bioinformatic analysis on 737 differentially expressed genes (Fig 5A and S1 Dataset). We extracted 1kb sequence 5' to the start codon of each gene and identified enriched sequence motifs by RSAT [50]. We then queried footprintDB for transcription factors that were predicted to bind these motifs [51], which yielded 42 transcription factors. An additional 10 transcription factors that were reported to regulate fat storage were considered [32,52,53]. Next, we curated a cherry-picked RNAi library of these 52 transcription factors in order to test if their depletion would modulate *alh-4(-)* phenotypes. To this end, we used the hypodermal peroxisome density and the fluorescence intensity of tagRFP::DAF-22, expressed from the endogenous *daf-22* locus, as reporters. Our functional genomic analysis revealed two nuclear receptors, NHR-49 and NHR-79 that are in part responsible for the transcriptional response in *alh-4(-)* worms. Our work corroborate with a prior study on NHR-49 target genes [54]. Specifically, peroxisome-related genes that were up-regulated in *alh-4(-)* worms were down-regulated in *nhr-49(-)* worms (S5A Fig).

To confirm our results based on RNAi, we introduced *nhr-49(lf)* or *nhr-79(lf)* mutant alleles into *alh-4(-)* worms. The loss of NHR-49 or NHR-79 significantly reduced peroxisome density and tagRFP::DAF-22 fluorescence intensity in *alh-4(-)* worms (Fig 5B–5E). We detected NHR-49 and NHR-79 expression in the intestine and hypodermis (S5C Fig). It is plausible that NHR-49 and NHR-79 act together in the hypodermis to regulate peroxisome proliferation and function in response to ALH-4 deficiency. Notably, NHR-79 was reported to interact with NHR-49 in yeast two-hybrid assays, albeit with much lower affinity than NHR-66 [32]. We verified these results by co-immunoprecipitation after expressing NHR-49 and NHR-79 or NHR-66 in mammalian cells (S5D Fig, compare lanes 9 and 10).

Next, we asked if NHR-49 and NHR-79 are also responsible for mounting a transcriptional response to ALH-4 deficiency in the intestine. The intestinal LDs in *nhr-49(lf)* or *nhr-79(lf)*

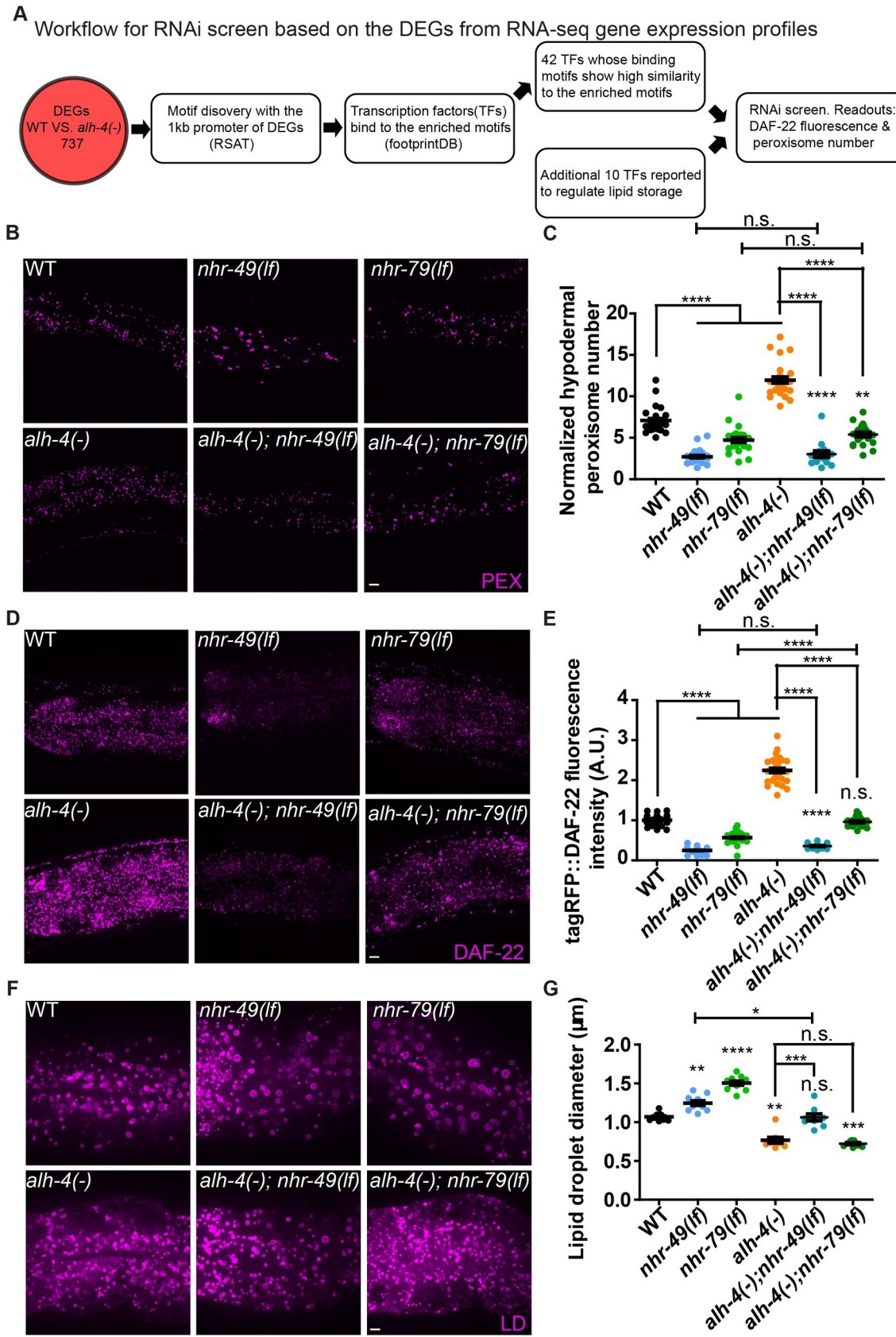

**Fig 5. Loss of *nhr-49* or *nhr-79* function suppresses ALH-4 deficiency.** (A) Workflow for RNAi screening based on differentially expressed genes (DEGs) identified by RNA sequencing of wild type (WT) and *alh-4(-)* worms. 737 genes were found to be differentially expressed with the threshold fold change >2 and p-value < 0.05(negative binomial test). (B) Representative images showing hypodermal peroxisomes labeled by tagRFP::PTS1 *(hjSi486)*, in WT, *nhr-49(lf)*, *nhr-79(lf)*, *alh-4(-)*, *alh-4(-)*; *nhr-49(lf)* and *alh-4(-)*; *nhr-79(lf)* L4 worms. An extra-chromosomal array that expressed functional GFP::ALH-4 *(hjEx25)* was used to propagate *alh-4(-)*; *nhr-49(lf)* worms. Based on the fluorescence of GFP::ALH-4, *alh-4(-)*; *nhr-49(lf)* worms with the extra-chromosomal array could be distinguished from those without it. (C) Quantification of hypodermal peroxisome number, n> 20 for each genotype from 3 independent experiments. (D) As in (B), but with animals expressing tagRFP::DAF-22 from the endogenous *daf-22* locus. (E) Quantification of tagRFP::DAF-22 fluorescence intensity. n >20 for each genotype from 3 independent experiments. (F) As in (B), but with animals expressing the lipid droplet marker DHS-3::mRuby *(hj200)*. (G) Quantification of lipid droplet size in the second intestinal segment. n = 10 for each genotype from 2 independent experiments. In each plot, mean ± SEM of each group is shown. For all statistical tests, *p < 0.05, **p < 0.01, ***p < 0.001, ****p < 0.0001, the actual p-value is displayed when p is between 0.05 and 0.1, n.s. (not significant) p > 0.1 (two-way ANOVA with Tukey's multiple comparisons test). PEX, peroxisome; LD, lipid droplet. Scale bar = 5μm.

mutant worms was significantly larger than those in wild type worms (Fig 5F–5G). However, the loss of NHR-49 but not NHR-79 partially corrected the intestinal LD size of *alh-4(-)* worms (Fig 5F–5G). Therefore, NHR-49 and NHR-79 appear to act independently in the intestine to control LD size.

## Intestinal ALH-4 regulates hypodermal peroxisome proliferation

The differential suppression of ALH-4 deficiency in *nhr-49(lf)* and *nhr-79(lf)* mutant worms suggests that tissue specific transcriptional programs may operate to sense cellular fatty aldehyde levels. If so, is organismal fatty aldehyde metabolism coordinated between tissues? To this end, we first sought to determine the site of action of ALH-4. We generated two single-copy transgenes that specifically expressed functional GFP::ALH-4 fusion proteins in the intestine or hypodermis of *alh-4(-)* worms (S6A Fig). The restoration of *alh-4* expression, as indicated by GFP fluorescence in the intestine, normalized the intestinal LD size and tagRFP::DAF-22 expression level (Fig 6A, 6C and 6D). Furthermore, the hypodermal peroxisome density and tagRFP::DAF-22 expression of these worms were corrected to the wild type level (Fig 6B–6D). In contrast, hypodermis-specific restoration of ALH-4 expression in *alh-4(-)* worms failed to normalize intestinal LD size, hypodermal peroxisome density, and tagRFP::DAF-22 expression level in both tissues (Fig 6A–6D). Our results indicate that intestinal ALH-4 regulates intestinal lipid storage cell-autonomously and hypodermal peroxisome proliferation cell-non-autonomously. In *alh-4(-)* worms, it is plausible that the alteration of cellular fatty aldehyde levels yields signals that act on the intestine and hypodermis in an autocrine and endocrine fashion, respectively.

## NHR-49 and NHR-79 act cell-autonomously in response to ALH-4 deficiency

Next, we determined the site of NHR-49 and NHR-79 action. We constructed transgenic worms that expressed GFP fusion of NHR-49 or NHR-79 specifically in the intestine or hypodermis from single copy transgenes, under the control of *vha-6* or *dpy-7* promoter, respectively. The tissue specific expression of fusion proteins was confirmed by the detection of primarily nuclear fluorescence signals in the appropriate tissues (S6B and S6C Fig). We found that NHR-49 acted cell-autonomously in the intestine and hypodermis. Accordingly, intestinal expression of NHR-49 normalized tagRFP::DAF-22 expression and LD size in the intestine but not hypodermis (Figs 6E, 6F and S6E). Hypodermal expression of NHR-49 specifically normalized hypodermal peroxisome density and tagRFP::DAF-22 expression and had no impact on mutant phenotypes in the intestine (Figs 6E, 6F and S6D). Similar observations were made for NHR-79: restoration of intestinal and hypodermal NHR-79 expression specifically suppressed mutant phenotypes in the respective tissues in a cell-autonomous manner

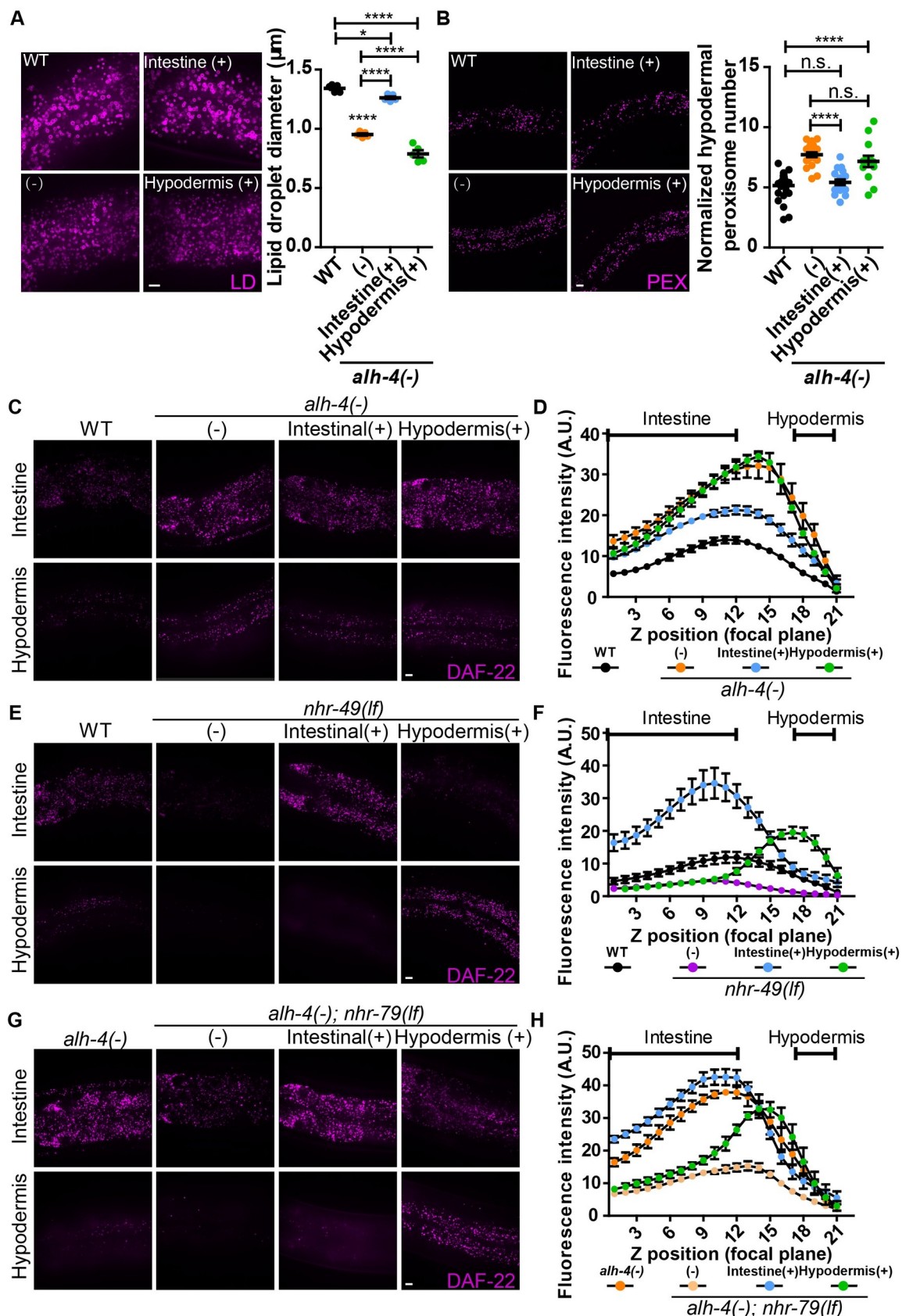

**Fig 6. Tissue specific rescue of *alh-4*, *nhr-49* and *nhr-79* mutant animals.** (A) Representative images showing intestinal lipid droplets labeled with DHS-3::mRuby (*hj200*) (left). Quantification of lipid droplet diameter in the second intestinal segment (right). n = 5 for each group. (B) Representative images showing hypodermal peroxisomes labeled with tagRFP::PTS1 (*hjSi486*) (left). Quantification of hypodermal peroxisomes normalized with the length of the worm in the field of view (right). n ≥ 13 for each group. (C) Representative images of Z-stack of 3 focal planes at 0.5μm interval showing tagRFP::DAF-22 in the intestine (top) and hypodermis (bottom) in wild type (WT), *alh-4(-)*, *alh-4(-); hjSi501[vha-6p::alh-4(+)]*, *alh-4(-); hjSi500[dpy-7p::alh-4(+)]* worms. (D) Quantification of fluorescence intensity of tagRFP::DAF-22 in each focal plane of Z stacks that encompass the intestine and hypodermis. n = 6 for each group. (E) As in (C), but with wild type (WT), *nhr-49(lf)*, *nhr-49(lf); hjSi531[vha-6p:: nhr-49(+)]*, *nhr-49(lf); hjSi537[dpy-7p:: nhr-49(+)]* worms. (F) As in (D), but with worms of specified genotypes. n = 5 for WT and n = 6 for other groups (G) As in (C), but with wild type (WT), *alh-4(-); nhr-79(lf)*, *alh-4(-); nhr-79(lf); hjSi553[vha-6p:: nhr-79(+)]*, *alh-4(-); nhr-79(lf); hjSi539[dpy-7p:: nhr-79(+)]* worms. (H) As in (D), but with worms of specified genotypes. n = 6 for each group. For all statistical tests, *$p < 0.05$, **$p < 0.01$, ***$p < 0.001$, ****$p < 0.0001$, the actual p-value is displayed when p is between 0.05 and 0.1, n.s. (not significant) $p > 0.1$ (one-way ANOVA with Tukey's multiple comparisons test). PEX, peroxisome; LD, lipid droplet. Scale bar = 5μm.

(Fig 6G, 6H and S6F). Taken together, we conclude that NHR-49 and NHR-79 act cell-autonomously in the intestine and hypodermis.

## NHR-49 functionally couples with NHR-79 to regulate peroxisome proliferation

Most nuclear receptors function as homodimers or heterodimers. NHR-49 partners with NHR-66, NHR-80 and NHR-13 to regulate distinct aspects of lipid metabolism [32]. NHR-49 also associated with NHR-79, based on co-immunoprecipitation (S5D Fig, lane 9) and yeast two-hybrid assays [32]. Therefore, we asked if NHR-49 and NHR-79 are functionally coupled to regulate fat storage and peroxisome proliferation in response to ALH-4 deficiency. First, we used tagRFP::DAF-22 fusion protein that was expressed from the endogenous *daf-22* locus as a reporter. In mutant worms that lacked NHR-49 or NHR-79, the expression of tagRFP::DAF-22 was reduced (Fig 7A and 7B). No further reduction was observed when both *nhr-49* and *nhr-79* were mutated (Fig 7A and 7B). Therefore, NHR-49 and NHR-79 appeared to act in concert to regulate *daf-22* expression. Next, we used an *nhr-49* gain-of-function (*gf*) allele to further substantiate our conclusion. There are three gain of function alleles of *nhr-49*: *nhr-49 (et7)*, *nhr-49(et8)* and *nhr-49(et13)* [36]. All alleles enhance the transcriptional activity of NHR-49 while *nhr-49(et13)* also causes de-repression of genes that are normally repressed by NHR-49 [55]. We chose *nhr-49(et8)* for our analysis, in order to focus on gene activation by NHR-49, in response to ALH-4 deficiency. We found that *daf-22* expression in *nhr-49(gf)* worms was elevated and such an effect was partially dependent on *nhr-79* (Fig 7A and 7B). NHR-49 and NHR-79 appeared to act in a similar manner to regulate hypodermal peroxisome density (Fig 7A and 7C). Next, we measured the intestinal lipid droplet size in *nhr-49* and *nhr-79* mutant worms. NHR-79 deficiency caused a larger increase in lipid droplet size than NHR-49 deficiency (Fig 7A and 7D). Interestingly, the lipid droplet size in *nhr-49(lf)* and *nhr-79(lf); nhr-49(lf)* mutant worms showed no significant difference. In addition, gain of NHR-49 function reduced the lipid droplet size of *nhr-79(lf)* worms (Fig 7A and 7D). Our results suggest that NHR-49 and NHR-79 do not regulate an identical set of genes to modulate lipid droplet size in the intestine. Taken together, NHR-49 and NHR-79 act together to regulate genes pertaining to peroxisome number and function. However, each of them may have additional functional partners to control fat storage in the intestine.

## NHR-49 is required for the growth and reproduction of ALH-4 deficient worms

Despite the accumulation of fatty aldehydes (Fig 1F) and presumably their toxic adducts, *alh-4 (-)* worms did not exhibit heightened sensitivity to oxidative stress (S4A and S4B Fig). We

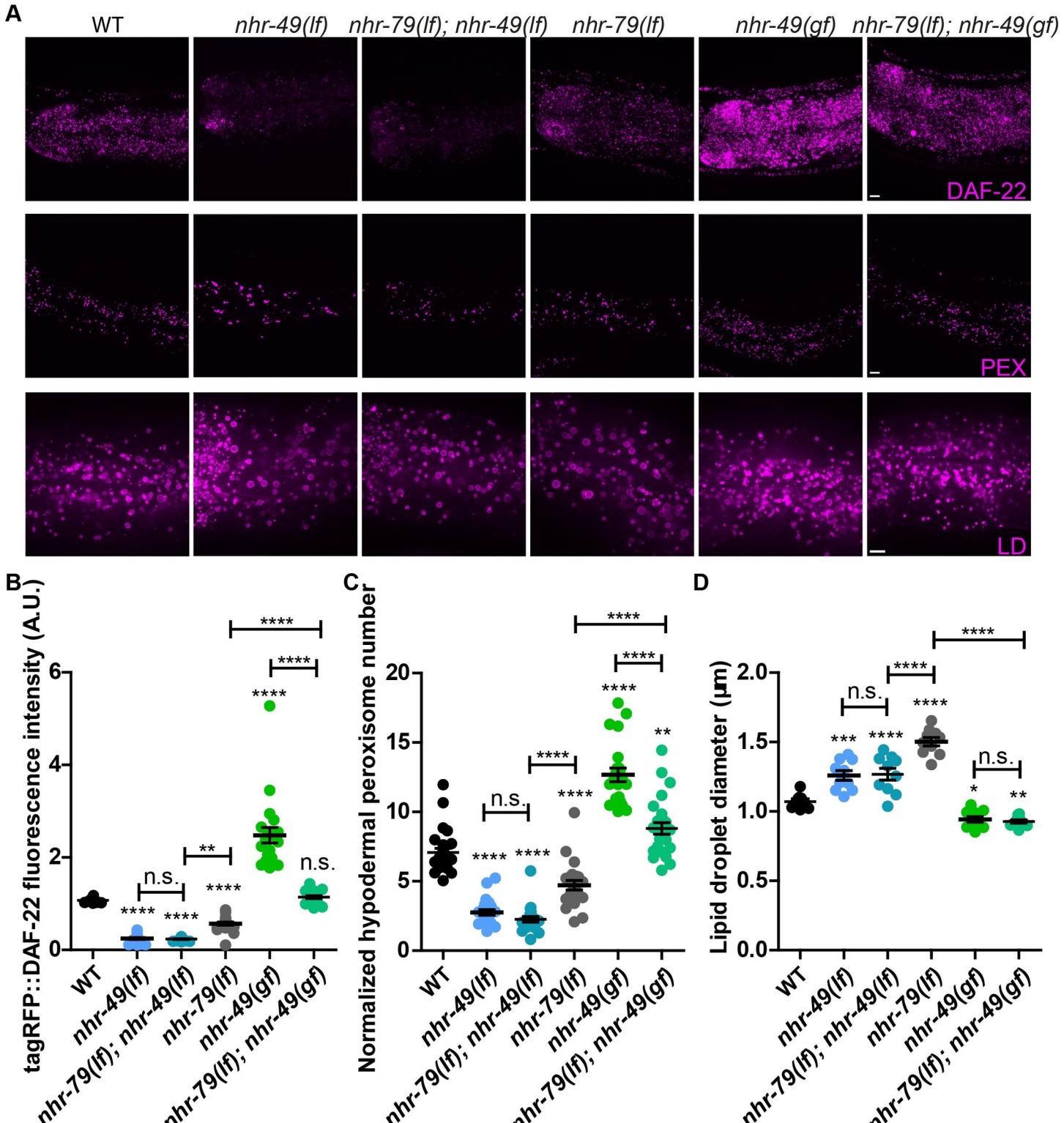

**Fig 7. Functional interaction of NHR-49 and NHR-79.** (A) Representative images showing the fluorescence of tagRFP::DAF-22 (*hj234*) (top), hypodermal peroxisomes labeled with tagRFP::PTS1(*hjSi486*) (middle), and intestinal lipid droplets labeled with DHS-3::mRuby (*hj200*) (bottom). (B) Quantification of fluorescence intensity of tagRFP::DAF-22. n > 20 for each group. (C) Quantification of hypodermal peroxisomes normalized with the length of the worm in the field of view. n > 20 for each group. (D) Quantification of lipid droplet size in the second intestinal segment. n = 10 for each group. In each plot, mean ± SEM of each group is shown. For all statistical tests, *p < 0.05, **p < 0.01, ***p < 0.001, ****p < 0.0001, the actual p-value is displayed when p is between 0.05 and 0.1, n.s. (not significant) p > 0.1 (two-way ANOVA with Tukey's multiple comparisons test). PEX, peroxisome; LD, lipid droplet. Scale bar = 5μm.

reasoned that NHR-49/NHR-79 dependent peroxisome proliferation could be regarded as a compensatory or hormetic response that maintained the health of *alh-4(-)* worms. To test this hypothesis, we used the number of progeny as an indicator of normal reproductive development. Both *alh-4(-)* and *nhr-49(lf)* mutant worms had a ~30% reduction of brood size (Fig 8A). However, almost all *alh-4(-); nhr-49(lf)* double mutant worms failed to produce live progeny after they developed into adults (Fig 8A). These sterile worms could be rescued by an extrachromosomal array that carried the wild type *alh-4* gene (Fig 8A). Furthermore, *alh-4(-); nhr-49(lf)* double mutant worms could be distinguished from *alh-4(-); nhr-49(lf);Ex[alh-4(+)]* and single mutant worms as they show obvious growth delay (Fig 8B). In contrast, the reproduction of *alh-4(-)* worms was not strictly dependent on NHR-79. Accordingly, *alh-4(-); nhr-79(lf)* mutant worms remained fertile, although their brood size was reduced in comparison to that of *alh-4(-)* or *nhr-79(lf)* single mutant worms (Fig 8A). We conclude that NHR-49 is a key factor that supports the development and reproduction of *alh-4* deficient worms, in part because of its broad role in regulating peroxisome proliferation and fat storage.

## Discussion

In this paper, we combined genetic, imaging, and gene expression profiling approaches to identify a peroxisome proliferative response that compensated for ALH-4 deficiency in *C. elegans* (Fig 8C). We observed concerted induction of genes that encode enzymes for peroxisomal fatty acid catabolism, which contributed to the reduction of neutral lipid storage in *alh-4* mutant worms. In addition, the upregulation of peroxisomal catalases in these worms conferred heightened resistance to exogenous hydrogen peroxide, an inducer of oxidative stress. Functional genomic analysis identified NHR-49 and NHR-79 as key transcription factors that were required for peroxisome proliferation in *alh-4* mutant worms. Interestingly, NHR-49 and NHR-79 acted cell autonomously in the hypodermis in response to ALH-4 deficiency in the intestine, thus hinting at an unknown endocrine signal that facilitates intestine to hypodermis communication. Our work revealed an unexpected homeostatic mechanism that alleviates fatty aldehyde dehydrogenase deficiency.

We originally cloned *alh-4* based on the observation that *alh-4* loss of function alleles suppressed fat accumulation and LD expansion in *daf-22* mutant worms, suggesting that *alh-4* acts downstream or in parallel of *daf-22*. Our subsequent molecular analysis clearly indicates the latter. Interestingly, the expression of *daf-22*, together with other peroxisomal β-oxidation enzymes, was induced in *alh-4* mutant worms. In contrast, the expression of genes encoding mitochondrial β-oxidation enzymes including acyl-CoA synthetases (ACSs), Enoyl-CoA Hydratases (ECHs), Carnitine Palmitoyl Transferases (CPTs), Acyl CoA Dehydrogenases (ACDHs) and thiolases were relatively unperturbed (S2 Dataset, sheet 2). Our results are consistent with the notion that *alh-4* deficiency specifically activates peroxisomal fatty acid β-oxidation pathways that act in parallel of the DAF-22 pathway. Therefore, enhanced fatty acid catabolism in the absence of ALH-4 prevented neutral fat storage in *daf-22* mutant worms.

By engineering worms that expressed a cytosolic form of ALH-4, we confirmed that membrane association is essential for its proper function. ALH-4 and mammalian FALDH/ALDH3A2 share the same mechanism of using divergent C-termini for tail-anchored organelle-specific association [10,11]. Therefore, it is conceivable that ALH-4 processes fatty aldehydes in situ as they arise in organelle membranes, thus necessitating the distribution of ALH-4 isoforms to multiple organelles. Surprisingly, we found that each of the ALH-4 isoforms was sufficient to rescue *alh-4(-)* mutant worms (S2 Fig and S1 Text). For example, the expression of ALH-4A, which associates with mitochondria and peroxisomes, relieved the requirement of ALH-4B and ALH-4C on the ER and LDs. It is formally possible that alternative aldehyde

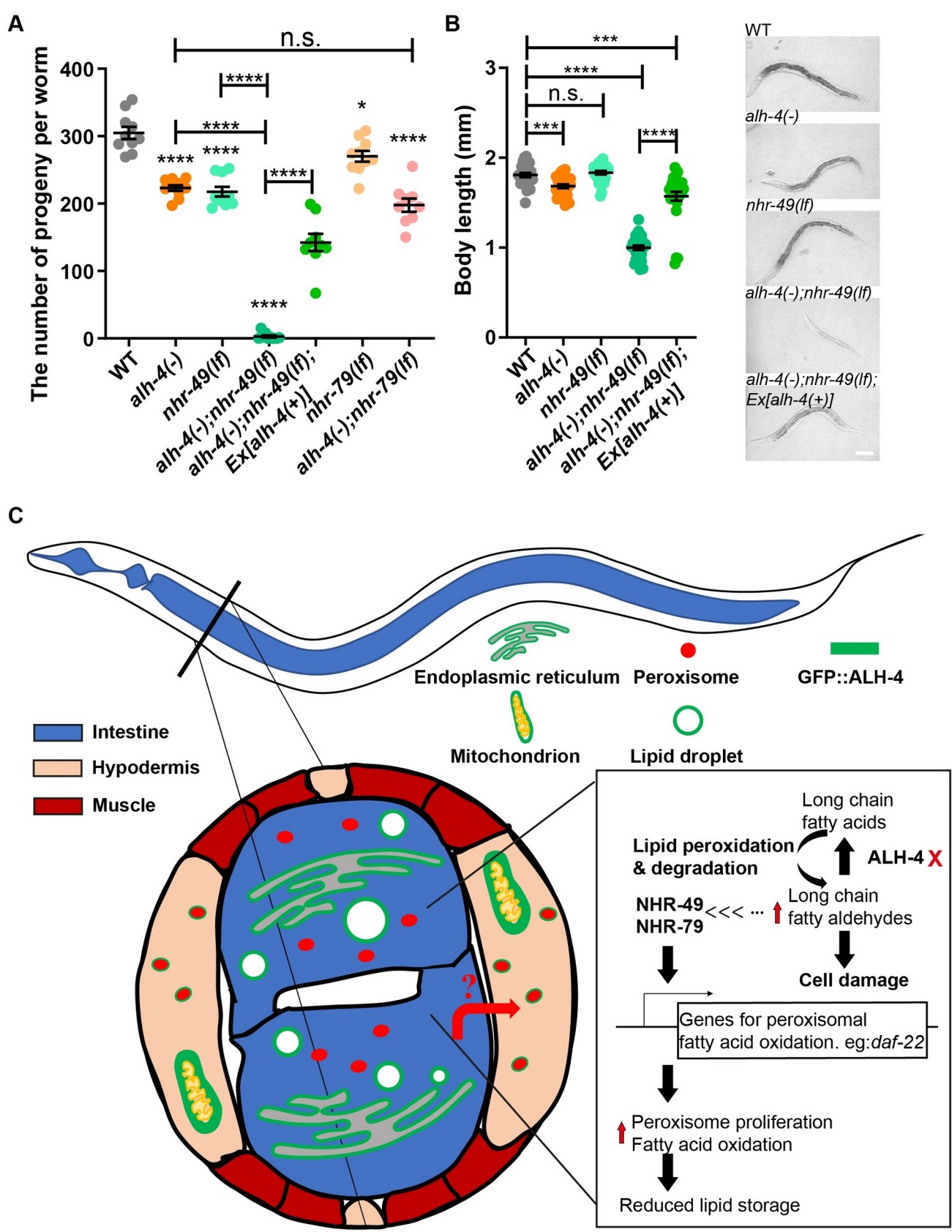

**Fig 8. NHR-49 is required for the growth and reproduction of *alh-4(-)* worms.** (A) Quantification of brood size of wild type (WT), mutant worms of indicated genotypes. *Ex[alh-4(+)]* represents *hjEx25(dpy-30p*::GFP:: *alh-4)*. n ≥ 8 for each group. In the plot, mean ± SEM of each group is shown. (B) Quantification of body length of wild type (WT), mutant worms of indicated genotypes 44 hours after hatching (left). The representative bright field images of worm body of indicated genotypes (right). Scale bar = 100μm. n = 26 for WT, *alh-4(-)* and *nhr-49(lf)*. n = 29 for *alh-4(-); nhr-49(lf)* and *alh-4 (-); nhr-49(lf); Ex[alh-4(+)]*. For statistical tests, *p < 0.05, **p < 0.01, ***p < 0.001, ****p < 0.0001, the actual p-value is displayed when p is between 0.05 and 0.1, n.s. (not significant) p > 0.1 (two-way ANOVA with Tukey's multiple comparisons test) with one exception: the comparison between *alh-4 (-); nhr-49(lf)* and *alh-4(-); nhr-49(lf); Ex[alh-4(+)]* was done with two-tailed unpaired Student's t-test. (C) A model for NHR-49/NHR-79 dependent peroxisome proliferation in response to ALH-4 deficiency.

dehydrogenases substituted for ALH-4 at the ER and LDs in this case. Nevertheless, our results raised the possibility that fatty aldehydes or their precursors, as part of a membrane lipid, could be transferred from one organelle to another, perhaps via inter-organelle contact sites.

The nuclear receptor NHR-49 partners with NHR-66, NHR-80 and NHR-13 to regulate distinct sets of target genes in lipid catabolism, desaturation and remodeling pathways [32]. Based on quantitative proteomic analysis, there is a good correlation between differentially expressed genes and their corresponding proteins in *nhr-49* knock-down worms [56]. Although NHR-49 has long been proposed as a functional ortholog of PPARα [31,34], the direct demonstration of its ability to promote peroxisome proliferation has been elusive. Here, we show that NHR-49, in partnership with NHR-79, is necessary for peroxisome proliferation in wild type and *alh-4* mutant worms. Interestingly, although NHR-49 and NHR-79 are both expressed in the intestine and hypodermis, the mRNA level of NHR-49 is >150 times higher than NHR-79 in wild type and *alh-4* mutant worms (S5B Fig). In comparison, the expression level of NHR-49 is 5 to 48 times higher than NHR-66, NHR-80 or NHR-13. This is consistent with the notion that NHR-49 serves as a common partner to multiple nuclear receptors in a tissue or context dependent manner. Accordingly, NHR-49 acts with NHR-79 to promote peroxisome proliferation in *alh-4* mutant worms. However, NHR-49 and NHR-79 regulate LD size in a divergent manner (Figs 5F, 5G and 7D).

It has been reported that phospholipids and their metabolites can serve as ligands for *C. elegans* NHRs [24,27,30]. For example, oleoylethanolamide, a PPARα agonist, binds NHR-80 to regulate *C. elegans* aging [24,57]. Although the polyunsaturated α-linolenic acid was reported to act via NHR-49 to extend *C. elegans* lifespan [38], no cognate ligand has been identified for NHR-49 or NHR-79 through direct binding assays. Interestingly, exogenous organic peroxide was reported to activate an NHR-49 dependent stress resistance program [54]. Taken together, it is tempting to speculate that excess fatty aldehydes or their metabolites in *alh-4* mutant worms may activate NHR-49 or NHR-79. Furthermore, the putative activators should be able to travel from the intestine to the hypodermis. This is because intestinal ALH-4 deficiency can trigger NHR-49/NHR-79 dependent peroxisome proliferation in the hypodermis. To substantiate our hypothesis, further analysis of metabolites that are enriched in *alh-4* mutant worms will be necessary.

The loss of FALDH/ALDH3A2 function causes Sjögren-Larsson Syndrome in humans [58]. The prominent phenotypes of spasticity and ichthyosis led to the proposal that the nervous system and skin are particularly reliant on FALDH for the metabolism of sphingolipids and the resolution of fatty aldehydes. The pathology of Sjögren-Larsson Syndrome has been partially recapitulated in FALDH/*Aldh3a2* knockout mice [59,60]. Detailed biochemical analysis of lipid species from skin and brain tissues of these knockout mice identified specific alteration in sphingolipid intermediates, which may contribute to keratinocytes and neuronal dysfunction. We note that ALH-4 deficient worms have excess fatty aldehydes or fatty alcohols (Figs 1F and S1G), similar to FALDH deficient fibroblasts [61]. However, we did not detect gross neuronal and epidermal defects based on available assays. This may be attributed to

fundamental differences in tissue architecture. For example, it has been hypothesized that myelin dysfunction contributes to neural symptoms in Sjögren-Larsson Syndrome [59]. However, *C. elegans* neurons are not myelinated. Taken together, instead of directly modeling the human disease, our work reveals a potentially conserved cellular mechanism that compensates for aldehyde dehydrogenase deficiency.

The concept of neurons-glia communication via lipid metabolites has been advanced in specific forms of neurodegeneration [62,63]. One therapeutic approach focused on the reduction of ROS, which are regarded as the upstream trigger of cell damage [62]. In principle, the modulation of latent, endogenous pathways may offer alternative therapeutic strategies. It is unclear if cell-cell communication also underlies the pathology of Sjögren-Larsson Syndrome. Our work indicates that an NHR-49 dependent, peroxisome proliferation and lipid detoxification program is required for the health and propagation of ALH-4 deficient worms. By analogy, fine-tuning of peroxisome proliferation or function via PPAR activation may be a probable strategy for treating Sjögren-Larsson Syndrome. Notably, PPARα induces FALDH expression [64] and the PPARα agonist Bezafibrate has been suggested as an agent for treating Sjögren-Larsson Syndrome patients with partially inactivated FALDH [65]. Additional natural and synthetic PPAR ligands have been actively tested as potential treatments for neurologic disorders [66,67]. It is conceivable that the same ligands can be repurposed to benefit Sjögren-Larsson Syndrome patients.

## Materials and methods

### *C. elegans* strains and maintenance

*C. elegans* strains are fed with *E. coli* OP50 on standard Nematode Growth Medium (NGM) plates at 20°C using standard protocol. Bristol N2 was used as the wild type. Strains carrying the following alleles were originally obtained from the *Caenorhabditis* Genetics Center (CGC): LG II, *daf-22(ok693)*; LG V, *nhr-79(gk20)*; LG I, *nhr-49(nr2041)*, *nhr-49(et8)*.

Mutant alleles first reported in this study are as follows.
LGV, *alh-4(hj27)*, *alh-4(hj28)*, *alh-4(hj29)*, *alh-4(hj221)*
Knock-in strains obtained with CRISPR-SEC [68]:
*alh-4(hj179) [GFP::TEV::loxP::3xFLAG::alh-4]*,
*dhs-3(hj200) [dhs-3::mRuby::TEV::3xFLAG]*,
*daf-22(hj234) [tagRFP::TEV::loxP::3xFLAG::daf-22]*,
*alh-4(hj261) [GFP::TEV::loxP::3XFLAG::alh-4(1-463aa)]*,
*alh-4 (hj264) [GFP::TEV::loxP::3xFLAG::alh-4(449-459aa deletion)]*,
*nhr-49(hj293) [nhr-49::GFP::TEV::3xHA]*.
Single-copy transgenes obtained with MosSCI [69] or CRISPR-SEC with a sgRNA targeting Mos1 (from Jihong Bai):
*hjSi112[vha-6p::3xFLAG::TEV::mRuby::F59A1.10 coding::let-858 3'UTR] IV, hjSi348[vha-6p::acs-22 cDNA::tagRFP::TEV::3xFLAG::let-858 3' UTR] IV, hjSi486[dpy-7p::tagRFP::PTS1::let-858 3' UTR] IV, hjSi489[dpy-30p::tomm-20(1-54aa)::mRuby::let-858 3' UTR] II, hjSi495 [dpy-30p::GFP::alh-4c(454-494aa)::let-858 3' UTR] I, hjSi497[dpy-30p::GFP::alh-4a(454-493aa)::let-858 3' UTR] I, hjSi499[dpy-30p::GFP::alh-4b(454-493aa)::let-858 3' UTR] I, hjSi500 [dpy-7p::GFP::TEV::loxP::3xFLAG::alh-4::dhs-28 3'UTR] I, hjSi501[vha-6p::GFP::TEV:: loxP::3xFLAG::alh-4::dhs-28 3'UTR] I, hjSi506[dpy-7p::GFP::PTS1:: let-858 3' UTR] IV, hjSi531 [vha-6p::nhr-49(a,b isoform cDNA with one short intron)::GFP::HA::dhs-28 3'UTR] II, hjSi537 [dpy-7p::nhr-49(a, b isoform cDNA with one short intron)::GFP::HA::dhs-28 3'UTR] II, hjSi539 [dpy-7p::GFP::3xFLAG::nhr-79a isoform(cDNA)::dhs-28 3'UTR] IV, hjSi548[vha-6p::mRuby:: PTS1::dhs-28 3' UTR] IV, hjSi553[vha-6p::GFP::3xFLAG::nhr-79a isoform(cDNA)::dhs-28*

3'UTR] II, hjSi554[dpy-30p::GFP::TEV::3xFLAG::alh-4a::let-858 3' UTR] I, hjSi555[dpy-30p:: GFP::TEV::3xFLAG::alh-4b::let-858 3' UTR] I, hjSi556[dpy-30p::GFP::TEV::3xFLAG::alh-4c:: let-858 3' UTR] I.

Extra-chromosomal array strains:

hjEx25[dpy-30p::GFP::TEV::3xFLAG::alh-4::let-858 3' UTR in pCFJ352]

hjEx26[nhr-79p::mRuby::3xFLAG::nhr-79::nhr-79 3'UTR; tub-1p::GFP::tub-1]

hjEx27, hjEx28 [vha-6p::GFP::TEV::loxP::3xFLAG::aldh3a2(1-463aa from cDNA NM_001031806.2)::alh-4(464aa-stop from N2 genomic DNA):: let-858 3' UTR; tub-1p::GFP:: tub-1]

All strains had been outcrossed with wild type N2 at least twice before being used.

## *C. elegans* genetic screen

The *daf-22(ok693)* mutant worms were mutagenized with Ethyl Methane Sulfonate (EMS) for suppressors of the expanded LD phenotype, as indicated by C1-BODIPY-C12 staining [39]. F2 progeny from 16340 F1 worms were screened using a fluorescence dissecting microscope. Mutant worms with reduced fluorescence and LD size were retained. Three mutant alleles (*hj27*, *hj28* and *hj29*) were mapped to *alh-4* using a single nucleotide polymorphism–based strategy with the Hawaiian *C. elegans* isolate CB4856 [70]. Molecular lesions of mutant alleles were determined by targeted Sanger sequencing. *hj27* encodes a G to A mutation that causes the substitution of Gly 415 to Glu. *hj28* and *hj29* are nonsense alleles. *hj28* carries a C to T substitution (Gln 301 stop), and *hj29* carries a G to A substitution (Trp 236 stop).

## RNA interference-based knockdown in *C. elegans*

The RNA interference (RNAi) constructs were transformed into a modified OP50 strain that is competent for RNAi [71]. Details of RNAi clones for transcription factors are available in S1 Dataset sheet 5. The target sequence in RNAi clones for *alh-4* and *atgl-1* were amplified by the following primer pairs: [5'-AGTTGTACTCATCATCTCCCCATG-3', 5'-TCCTCGCCATAA AATTCGTTC-3'] and [5'-TGGAGCATCTTCGTCTTCCTC-3', 5'-AAAATCATATAAAT TTCAGC-3'], respectively from genomic DNA. OP50 RNAi clones were streaked on LB plates with 100μg/mL ampicillin and tetracycline and cultured with 2xYT with 100μg/mL ampicillin overnight. Fresh liquid culture was seeded onto NGM plates for RNAi. The seeded plates were incubated at room temperature for 1 day and 10 L4 worms were transferred on the seeded RNAi plates. When worms reached day 1 adult, they were transferred to new RNAi plates to lay eggs for overnight and then removed from the plates. F1 worms at L4 stage were used for imaging.

## Hydrogen peroxide treatment

Synchronization was done by collecting L1 worms that were hatched within a 2-hour window. 200 L1 worms were transferred to 60mm NGM plates seeded with OP50. After 48 hours, L4 worms were collected with M9, transferred to 1.5mL Eppendorf tubes and washed three times with M9 buffer and then aliquoted into 2 tubes, one for experimental group and the other for negative control. For experimental groups, M9 buffer with hydrogen peroxide was added to worms (final concentration = 0.75mM). For negative control, M9 without hydrogen peroxide was added. Worms were aliquoted into 24 well plates, on average about 20 worms per well and 3 wells for worms collected from 1 plate and then incubated at 20°C for 4 hours. To stop the hydrogen peroxide treatment, OP50 bacteria liquid culture was added. Worms were scored after a recovery period of 30 minutes. Worms that did not show detectable movement were categorized as paralyzed. Paralysis rate = (number of the paralyzed worms / total number of

worms) $_{\text{experimental group}}$—(number of the paralyzed worms / total number of worms) $_{\text{negative}}$
$_{\text{control group}}$.

## Paraquat treatment

The procedure for paraquat treatment is the same as that for hydrogen peroxide treatment except that the worms were incubated in M9 with 100mM paraquat instead of hydrogen peroxide.

## Brood size

Individual L4 worms were introduced onto NGM plates seeded with OP50 and transferred to new plates every day until they stopped egg laying. 1.5 days after the mother worms were removed, the number of larvae on each plate was counted. The total number of progeny from each adult worm was calculated and recorded as the brood size.

## Growth rate

50–100 adult worms were transferred to NGM plate seeded with OP50 to lay eggs for 1 hour and then 50 eggs were transferred to each new plate. 70 hours after egg laying, larvae were classified as early L4, mid L4, the stage between L4 and adult or adult stage based on the valval morphology.

## Pharyngeal pumping rate

L4 worms were transferred onto new NGM plates seeded with OP50, 2 worms per plate. After 24 hours, the pharyngeal pumping of 1-day old adults was video recorded on a dissecting microscope for at least 1min at room temperature. The videos were played with the PotPlayer at 0.3X speed for precise counting of pharyngeal muscle contractions per min.

## Fluorescence imaging of *C. elegans*

Fluorescence images were obtained on a spinning disk confocal microscope (AxioObeserver Z1, Carl Zeiss) equipped with a Neo sCMOS camera (Andor) controlled by the iQ3 software (Andor). The excitation lasers for GFP, tagRFP/ mRuby were 488nm and 561nm and the emission filters were 500-550nm and 580.5–653.5nm, respectively. The images were imported to Imaris 8 (Bitplane) for image processing and analysis and exported as files in ims format. The ims files were further imported into ImageJ for analysis.

## The quantification of lipid droplet size

Lipid droplets in the intestinal cells were labeled with fluorescent protein tagged resident lipid droplet proteins. Fluorescence images of L4 larvae were acquired on a spinning disk confocal microscope with a 100x objective. For each worm, a Z-stack was taken at 0.5μm interval and 10μm thickness (21 layers) starting at the layer between the intestinal cell and the hypodermis. From each stack, 10 continuous focal planes were chosen to construct a 3D image. The Z-stack of bright field images was used to determine the outline of the intestinal cells. The lipid droplets in the second intestinal segment were labeled manually with the spot function in the Imaris 8 (Bitplane) and the diameter of each lipid droplet were exported and used to calculate the average diameter of lipid droplets for each worm. When calculating the average lipid droplet diameter, lipid droplets with diameter < 0.5 micrometer were excluded because the diffraction limit of light microscopy preluded their accurate measurement.

## The quantification of peroxisome number

Peroxisomes were labeled with fluorescent protein tagged with a PTS1 signal peptide. Fluorescence images of L4 larvae were acquired on a spinning disk confocal microscope with a 63x objective. For each worm, a Z-stack of its anterior region without the head was taken at 0.5μm interval and 4.5μm thickness, focusing on the tissue of interest. A Z-stack of bright field images was also obtained. Z-stack images were imported to Imaris 8 (Bitplane) for 3D reconstruction. The peroxisomes were automatically detected by the spot function with the following algorithm: Enable Region of interest = false; Enable region growing = true; Source channel index = 2; Estimated Diameter = 0.5; Background Subtraction = true; 'quality' above 1.000; Region Growing type = Local contrast; Region Growing automatic Threshold = false; Region Growing manual Threshold = 9; Region Growing Diameter = diameter from volume; Create region channel = false. Spots representing mis-labeled peroxisomes or the peroxisomes not in the tissue of interest were manually deleted. For the quantification of the hypodermal peroxisomes, only the peroxisomes in the hypodermis close to the coverslip were counted. With the statistics function, the number of peroxisomes can be obtained. The length of the worm body in the view was measured based the bright field images. The peroxisome number was normalized by the length of the worm body in the field of view.

## The quantification of DAF-22 fluorescence intensity

To measure the expression level of DAF-22, the coding sequencing of tagRFP was inserted at the 5' end of the endogenous *daf-22* gene by CRISPR, to yield transgenic worms that expressed tagRFP::DAF-22 fusion protein at the endogenous level. Fluorescence images of L4 larvae were acquired on a spinning disk confocal microscope with a 63x objective. For each worm, a Z-stack was taken with 0.5μm interval and 10μm thickness starting at the layer closest to the coverslip with obvious fluorescence signals (i.e. hypodermis) and ending at intestinal layers. A Z-stack of bright field images was also obtained. The field of view included the anterior region of each worm without the head. Using Imaris 8 (Bitplane), the outline of the worm body was manually drawn based on bright field images, the surface function was used to create a worm surface which colocalizes with the surface of the worm. Similarly, a background surface was manually drawn and created in the region of the field without the worm. The average intensity of tagRFP::DAF-22 is the average intensity inside of the worm body surface minus that of the background surface. The normalization of fluorescence intensity was done with the average intensity of WT worms that were imaged at the same time.

## RNA-seq of *C. elegans*

Three biological replicates of N2 and *alh-4(-)* were synchronized and grown to L4 stage. Sequencing library preparation was done according to a published method [72] with the modifications: the number of worms for each replicate and the reagent for worm lysis were scaled up by 10 fold and the worms were broken with a mini-pestle in a 1.5ml disposable tube on ice instead of cutting. The libraries were pooled and sequenced with the Illumina NextSeq 500 system. The reads were mapped to ce10 genome assembly. The ce10 Illumina iGenome gene annotation was used as a reference for transcripts identification. The downstream gene expression level analysis was done based on the published method [73] to get FPKM (Fragments Per Kilobase Million) values for differential gene expression analysis.

## Motif analysis

The list of the differentially expressed genes between WT and *alh-4(-)* is obtained with the threshold fold change >2 and p-value < 0.05 (negative binomial test). -1000 to -1 sequence

relative to the start codon of differentially expressed genes (DEGs) were obtained for motif discovery to identify enriched motifs with RSAtool [50]. FootprintDB was used to identify transcription factors that are predicted to bind the enriched motifs [51].

## Gene ontology and pathway analysis

The analysis was done with gProfiler [47] using the list of DEGs between WT and *alh-4(-)* as the input and choosing *Caenorhabditis elegans* as the organism.

## SRS for lipid content measurement

To measure intestinal lipid content, $CH_2$ signal was imaged at 2863.5 $cm^{-1}$. Live worms at L4 stage were mounted on 8% agarose pad in PBS buffer with 0.2mM levamisole. The focal plane with maximal SRS signals was determined with a 20x air objective (Plan-Apochromat, 0.8 NA, Zeiss). The quantification of the SRS signal was done following a published protocol [74]. In short, the average SRS signal intensity in the intestine region per pixel was calculated with the background signal subtracted. The signal intensity for each worm was normalized with the average value of the WT group for the data display.

## Lipid extraction

Lipid were extracted according to a modified version of the methyl-tertiary-butyl ether (MTBE) extraction originally developed by Matyash *et al* [75,76]. 2000 synchronized L4 worms were collected into organic solvent resistant Eppendorf tubes and suspended in 250μL precooled methanol. Subsequently, 875 μL MTBE was added. Worms were lysed with ice cold ultrasonic bath for 30 minutes. Phase separation was induced by adding 210 μL water followed by sonication for 15 minutes. The organic supernatant was collected after centrifugation at 4˚C. Re-extraction of the lower phase was done by adding additional 325 μL MTBE. The organic supernatant was collected and combined with those collected previously and evaporated with under a stream of nitrogen at room temperature and then stored at -80˚C. Before analysis, lipids were dissolved in 200 μL acetonitrile / isopropanol / water (65/30/5, v/v/v) and a 100 μL aliquot was used for LC-MS analysis per sample.

## Liquid chromatography

Lipid separation was done with the Bruker Elute UPLC system (Brucker) on a Bruker intensity Solo 2 C18 column (100 x 2.1 mm ID, 1.7 μm particle size) (Brucker). The column temperature was held at 55˚C and the lipids were separated with a gradient from eluent A (acetonitrile / water (60:40, 10 mM NH4 Formate, 0.1% formic acid (v/v)) to eluent B (isopropanol / acetonitrile (90:10, 10 mM NH4 Formate, 0.1% formic acid (v/v)) with a flow rate of 400 μL/min. Gradient conditions: After an isocratic step of 40% B for 2 min, then 43% B for 0.1 min, 50% B for 10min, 54% B for 0.1min, 70% B for 6min and 99% B for 0.1 min. After returning to initial conditions, the column was re-equilibrated for 2 min with the starting conditions.

## Trapped ion mobility-PASEF mass spectrometry

Output from liquid chromatography was injected into a trapped ion mobility-quadrupole time-of-flight mass spectrometer (timsTOF Pro MS) (Brucker) through an electrospray ion source. The mass spectrometry was powered by the Parallel Accumulation Serial Fragmentation (PASEF). Analysis was performed twice for each sample in both positive and negative electrospray modes. The parameter was modified based on a published method [77]. The mass

spectra were scanned over the range of m/z 100 to m/z 1350 while the ion mobility was recorded from 0.55 Vs cm$^2$ to 1.90 Vs cm$^2$. Precursors within ± 1 Th were isolated for MS/MS acquisition. 10 eV was used as collision energy for fragmentation. The acquisition cycle was composed of one full TIMS-MS scan and 2 PASEF MS/MS scans, taking 0.31s. Precursor ions with an intensity within the range of 100–4000 counts was rescheduled. PASEF exclusion release Time was set to 0.1 min. IMS ion charge control was $5 \times 10^6$. The calibration of TIMS was done with the Agilent ESI LC/MS tuning mix (Agilent Technology). The selected ions for the calibration of positive modes were: [m/z, 1/K0: (322.0481, 0.7318 Vs cm$^{-2}$), (622.0289, 0.9848 Vs cm$^{-2}$), (922.0097, 1.1895 Vs cm$^{-2}$), (1221,9906, 1.3820 Vs cm$^{-2}$)] while those for negative modes were: [m/z, 1/K0: (601.9790, 0.8782 Vs cm$^{-2}$), (1033.9882, 1.2526 Vs cm$^{-2}$), (1333.9690, 1.4016 Vs cm$^{-2}$), (1633.9498, 1.5731 Vs cm$^{-2}$)]. For negative mode, only mass was calibrated but not the mobility.

## Lipid annotation and statistical analysis

The mass spectrometry data was analyzed with MetaboScape version 5.0 (Bruker Daltonics), which extracts retention time, m/z, collision cross section and intensity for each chemical detected while assigning MS/MS spectra to it. After obtaining the merged bucket table from both the positive and negative modes, detected chemicals were firstly annotated with Spectral Libraries: MSDIAL- Tandem Mass Spectral Atlas-VS68-pos and MSDIAL- Tandem Mass Spectral Atlas-VS68-neg. and then annotated with analyte lists of fatty aldehydes [FA06], fatty alcohols [FA05], and fatty Acids and Conjugates [FA01], sphingolipids [SP] from LIPID MAPS (https://www.lipidmaps.org/). Finally, entries in the bucket table were annotated with SmartFormula. The intensity was normalized with probabilistic quotient normalization [78]. The following lipid categories were searched: Triacylglycerol (TAG), Phosphatidylethanol-amine (PE), Phosphatidylcholine (PC), Phosphatidylserine (PS), Phosphoinositide (PI), ether linked phospholipid, sphingolipid, sphingomyelin, fatty aldehyde/fatty acid (MS information was insufficient to distinguish fatty aldehyde and fatty alcohol), fatty acids and diacylglycerol (DAG). The average intensity of the two technical replicates (0 value was considered as missing value (NA) and therefore ignored) for each chemical detected in that category were summed up to get the total amount for the corresponding categories.

## Immunoprecipitation and western blotting

Human embryonic kidney 293 cells cultured in 6-well plates were transfected with ~3 μg plas-mid per well using lipofectamine 2000 (Invitrogen). The following plasmids were used with their predicted molecular weight noted after their name:

pcDNA5FRT_2xHA::Venus, 29.5 KDa
pcDNA5FRT_FLAG::NHR-49B (87-476aa), 44.1KDa
pcDNA5FRT_2xHA::NHR-66A (191-577aa), 46.5 KDa
pcDNA5FRT_2xHA::NHR-79A (156-463aa), 38.4 KDa

Cells were collected 2 days after transfection in cell lysis buffer (150mM NaCl, 50mM Tris-HCl with 1% NP-40, pH7.8, 1X protease inhibitors [Complete; Roche], 20mM NaF and 1mM PMSF) and lysed by passing through 25G gauge needle 10 times. The supernatant of the cell lysate was incubated with anti-FLAG M2 Magnetic Beads (Sigma-Aldrich). Immunoprecipita-tion and washing were performed in PBS buffer with 1X protease inhibitors [Complete; Roche], 20mM NaF and 1mM PMSF. Proteins were eluted with 2 × SDS loading buffer and boiled at 70˚C for 10mins before SDS-PAGE. For Western blotting, anti-FLAG (F1804, Sigma) and anti-HA (11867423, Roche) antibodies were used for detecting FLAG-tagged and HA-tagged proteins, respectively.

## Statistical analysis

Statistical analysis was performed in GraphPad Prism or Microsoft Excel and the type of statistical test applied was indicated in the figure legend.

## Supporting information

**S1 Fig. Phenotypic characterization of *alh-4* mutant worms.** (A) Schematic representation of the *alh-4* gene structure. Black boxes, exons; white boxes, untranslated region; line, splicing. The mutation sites in *hj28* and *hj29* are indicated by red triangle and yellow triangle, respectively. The single nucleotide substitution in both alleles result in a premature stop codon. The mutation site in *hj27* is indicated by black triangle and the single nucleotide substitution results in a missense mutation Gly 415 Glu. The deletion region in *alh-4* in *hj221* is indicated with the blue line. *alh-4(hj221)* is labeled as *alh-4(-)* in all figures. Scale bar = 100bp. (B) Representative images of lipid droplets in the second intestinal segment of L4 worms. Lipid droplets were labeled with mRuby::DGAT-2 (*hjSi112*). Scale bar = 5μm. (C) Quantification of number of progeny of wild type (WT) and mutant worms of indicated genotypes. Two-way ANOVA with Tukey's multiple comparisons test was applied. (D) Percentage of 50 synchronized worms in the young L4, middle L4, between L4 and adult or adult developmental stage observed at 70 hours after egg laying. (E) Quantification of the pharyngeal pumping rate of WT, *alh-4(hj29)* and *alh-4(hj221)* worms. n = 10 for each group. Two-tailed unpaired Student's t-test was applied. (F) Frequency distribution of LD diameter of wild type (WT) and mutant worms of indicated genotypes. Total number of the lipid droplet measured: WT = 1055, *alh-4(hj29)* = 1628, *daf-22(ok693)* = 216, *alh-4(hj29); daf-22(ok693)* = 943. (G) Quantification of indicated fatty aldehydes and fatty alcohols in wild type (WT) and *alh-4(-)* worms with LC-MS. A significant increase in multiple species of fatty aldehyde/fatty alcohol was observed in *alh-4(-)* worms in comparison with WT worms. n = 3 for each group except 0 value was considered as missing value (NA) and therefore ignored. Each dot represents the average value of two technical replicates of each biological sample. Two-tailed unpaired Student's t-test was applied. For all plots, mean ± SEM of each group is shown. For all statistical tests, $^*p < 0.05$, $^{**}p < 0.01$, $^{***}p < 0.001$, $^{****}p < 0.0001$, the actual p-value is displayed when p is between 0.05 and 0.1, n.s. (not significant) p > 0.1.
(TIF)

**S2 Fig. Subcellular localization of ALH-4 and transgenic rescue of *alh-4(-)* worms.** (A) Schematic representation of the structure of transgenes expressing GFP tagged ALH-4 isoforms. The magenta box indicates isoform specific sequence. ALH-4A isoform: AAATATGCT CGCAATCTTCATTGA encoding KYARNLH* ALH-4B isoform: TTCATTTTCCGCTTCTC GGCATAA encoding FIFRFSA*. ALH-4C isoform: GTATGCAGAGGAAAATCAGCTCAA TAG encoding VCRGKSAQ*. Scale bar = 100bp. (B—E) GFP::ALH-4A (*hjSi554*) colocalized with mitochondria labeled with TOMM-20N::mRuby (*hjSi489*) and peroxisomes labeled with tagRFP::DAF-22 (*hj234*) in the intestine and tagRFP::PTS1 (*hjSi486*) in the hypodermis. (F) GFP::ALH-4B (*hjSi555*) colocalized with the ER labeled with ACS-22::tagRFP (*hjSi348*) in the intestine. (G) No distinct pattern of GFP::ALH-4B was detected in the hypodermis. (H—I) In the intestine, GFP::ALH-4C (*hjSi556*) colocalized with lipid droplets labeled with DHS-3:: mRuby (*hj200*) and mitochondria labeled with TOMM-20N::mRuby (*hjSi489*). (J) In the hypodermis, GFP::ALH-4C colocalized with mitochondria labeled with TOMM-20N::mRuby (*hjSi489*). (K) Table summarizing the subcellular localization of different ALH-4 isoforms in the intestine and hypodermis. (L) Schematic representation of the structure of transgenes expressing GFP fused with the C-terminus of ALH-4 isoforms. Scale bar = 100bp. (M)

Representative images showing the subcellular localization of GFP fused with C-terminus of ALH-4 isoforms in the intestine (top) and in the hypodermis (bottom). (N) Table summarizing the results shown in (M). (O) Transgenic rescue of *alh-4(-)* worms with specific ALH-4 isoforms. Representative images showing intestinal lipid droplets labeled with DHS-3::mRuby (*hj200*). (P) As in (O), but with representative images showing the hypodermal peroxisomes labeled with tagRFP::PTS1(*hjSi486*). (Q) Quantification of intestinal lipid droplet diameter. n = 5 for each group. (R) Quantification the hypodermal peroxisome number normalized with the length of the worm in the field of view. n ≥ 6 for each group. In each plot, mean ± SEM of each group is shown. $^*$p < 0.05, $^{**}$p < 0.01, $^{***}$p < 0.001, $^{****}$p < 0.0001, the actual p-value is displayed when p is between 0.05 and 0.1, n.s. (not significant) p > 0.1 (two-way ANOVA with Tukey's multiple comparisons test). PEX, peroxisome; LD, lipid droplet. Scale bar = 5μm. (TIF)

**S3 Fig. Engineered mutations for the disruption of the membrane anchor or gate-keeper region of ALH-4.** (A) Phylogram of human ALDHs and *C. elegans* ALHs. ALH-4 is most closely related to class III ALDHs based on their primary amino acid sequences. The length of the branch shows the genetic distance. For human proteins with isoforms, the amino acid sequence of the longest isoform was used for sequence comparison. (B) Sequence alignment of human and mouse ALDH3A2 and *C. elegans* ALH-4A (with Clustal X2). Residue color scheme: hydrophobic (A, L, M, P, I, F, W, V), red; polar (N, C, Q, T, Y, G, S), green; negative charged (D, E), blue; positive charged (H, R, K), yellow. Alignment quality display: '*' for a single, fully conserved residue; ':' for fully conserved of some "strong groups"; '.' for fully conserved of some "weak groups"; Corresponding regions are in grey areas. (C) Schematic representation of *gfp::alh-4(Δgk)* (*hj264*) and *gfp::alh-4(ΔC)* (*hj261*), which had the gate-keeper region and C-terminal membrane anchor deleted, respectively. The deleted region is indicated. TMD, transmembrane domain. Scale bar = 100bp. (TIF)

**S4 Fig. Phenotypic analysis of *alh-4(-)* worms.** (A) Scatter plot showing the percentage of worms paralyzed after being treated with 100mM Paraquat for 6 hours. Three independent experiments were carried out and each with three biological replicates. The total number of worms measured in 9 biological replicates were 4, 13, 7 for WT and 34, 8,12 for *alh-4(-)* in the first run; 8, 10, 36 for WT, 19, 12, 13 for *alh-4(-)* in the second run and 14, 21, 19 for WT and 25, 10, 11 for *alh-4(-)* in the third run. Data points are color-coded to indicate results from specific experiments. (B) As in (A) but showing the percentage of worms paralyzed after 4 hours treatment with 0.75mM hydrogen peroxide in M9 buffer. The total number of worms measured in 9 biological replicates were 41, 27, 35 for WT and 32, 28, 33 for *alh-4(-)* in the first run; 16, 21, 22 for WT and 17, 24, 35 for *alh-4(-)* in the second run and 25, 28, 27 for WT and 29, 28, 29 for *alh-4(-)* in the third run. (C) Representative images (Z-stack, 0.5 μm intervals, 10 slices) showing the intestinal peroxisomes labeled with mRuby::PTS1 (*hjSi548*) (top). Quantification of intestinal peroxisome number (bottom). n = 11 each group. (D-E) Differential effects of *atgl-1* knockdown by RNAi on peroxisome- and LD-related phenotypes of *alh-4(-)* worms. (D) Representative images of LDs labeled with DHS-3::mRuby (*hj200*) in the second intestinal segment. (E) Quantification of the average LD diameter in the second intestinal segment. $^*$p < 0.05, $^{**}$p < 0.01, $^{***}$p < 0.001, $^{****}$p < 0.0001, the actual p-value is displayed when p is between 0.05 and 0.1, n.s. (not significant) p > 0.1 (two-way ANOVA with Tukey's multiple comparisons test). (F) Representative images showing tagRFP::DAF-22 (*hj234*) in *alh-4(-)* worms that were subject to control (L4440) or *atgl-1* RNAi (left). Quantification of tagRFP::DAF-22 fluorescence (right). n = 8 for each group. (G) As in (F), but with hypodermal peroxisomes labeled with tagRFP::PTS1(*hjsi486*) (left). Quantification of peroxisome number

normalized with length of the worms in the field (right). n = 8 for each group. PEX, peroxisome. Scale bar = 5μm. In each plot, mean ± SEM of each group is shown. (A-C, F-G), $^*p < 0.05$, $^{**}p < 0.01$, $^{***}p < 0.001$, $^{****}p < 0.0001$, the actual p-value is displayed when p is between 0.05 and 0.1, n.s. (not significant) $p > 0.1$ (two-tailed unpaired Student's t-test). (TIF)

**S5 Fig. Transcriptomic, imaging and biochemical analysis of NHR-49 and NHR-79.** (A) Heatmap showing the up-regulated genes (fold change > 2 and p-value < 0.05, negative binomial test) in *alh-4(-)* worms in the peroxisome pathway are generally down-regulated in *nhr-49(lf)* worms. FPKM value is converted to Z-scores. High expression is indicated with red, while low expression is indicated with blue. The data for the WT and *nhr-49* groups on the right was extracted from GSE107799 at Gene Expression Omnibus. (B) Histogram showing the mRNA levels of *nhr-49*, *nhr-79*, *nhr-80*, *nhr-66* and *nhr-13*, from RNA sequencing of wild type (WT) and *alh-4(-)* worms. n = 3 for each group. In the plot, mean ± SEM of each group is shown. $^*p < 0.05$, $^{**}p < 0.01$, $^{***}p < 0.001$, $^{****}p < 0.0001$, the actual p-value is displayed when p is between 0.05 and 0.1, n.s. (not significant) $p > 0.1$ (two-tailed unpaired Student's t-test). (C) Visualization of NHR-49::GFP (*hj293*) from the endogenous locus (top) and the visualization of mRuby::NHR-79 (*hjEx26[nhr-79p::nhr-79]* (bottom). Both NHR-49::GFP and mRuby::NHR-79 could be detected in the nuclei of intestinal and hypodermal cells. Scale bar = 5μm. (D) Co-immunoprecipitation of NHR-49 and NHR-79. The predicted ligand binding domain of NHR-49B(87-476aa), NHR-79A(156-463aa) and NHR-66A(191-577aa) were transiently expressed as FLAG- or HA-tagged proteins in HEK293T cells. HA::Venus was expressed as a negative control. At least two experiments with independent biological samples were performed. A representative blot is shown. 20% input of each sample was loaded. (TIF)

**S6 Fig. Tissue specific rescue of *alh-4(-)*, *nhr-49(lf)* and *nhr-79(lf)* worms.** (A) Representative images showing the intestinal specific expression of GFP::ALH-4 in worms carrying *hjSi501*(left) and the hypodermal specific expression of GFP::ALH-4 in worms *hjSi500* (right). (B) Representative images showing the intestinal specific expression of NHR-49A/B::GFP in worms carrying *hjSi531* (left) and the hypodermal specific expression of NHR-49A/B::GFP in worms carrying *hjSi537* (right). (C) Representative images showing the intestinal specific expression of GFP::NHR-79A in worms carrying *hjSi553*(left) and the hypodermal specific expression of GFP::NHR-79A in worms carrying *hjSi539* (right). (D) Representative images showing hypodermal peroxisomes labeled with tagRFP::PTS1 (*hjSi486*) (top). Quantification of hypodermal peroxisome number in wild type (WT) and mutant worms of indicated genotypes. n ≥ 6 for each group. (E) As in (D), except with intestinal lipid droplets labeled with DHS-3::mRuby (*hj200*) (top). Quantification of lipid droplets in the second intestinal segment (bottom). n = 6 for each group. (F) As in (D), except with WT and *alh-4(-)* and *nhr-79(-)* mutant worms of indicated genotypes. Quantification of hypodermal peroxisome number (bottom). n ≥ 6 for each group. In each plot, mean ± SEM of each group is shown. For all statistical tests, $^*p < 0.05$, $^{**}p < 0.01$, $^{***}p < 0.001$, $^{****}p < 0.0001$, the actual p-value is displayed when p is between 0.05 and 0.1, n.s. (not significant) $p > 0.1$ (one-way ANOVA with Tukey's multiple comparisons test). In all plots, mean ± SEM of each group is shown. PEX, peroxisome; LD, lipid droplet. Scale bar = 5μm. (TIF)

**S1 Dataset. Gene expression profiles of WT and *alh-4(-)* obtained by RNA-sequencing (sheet 1) and the down-stream analysis.** 1. Differentially expressed genes (DEGs) analysis (Sheet 2); 2. Gene ontology and Kyoto Encyclopedia of Genes and Genomes (KEGG) pathway

analysis (Sheet 3); 3. Motif discovery of the promoter region of DEGs (Sheet 4); 4. The transcription factor candidates and RNAi results (Sheet 5). The RNA-sequencing dataset has been deposited in NCBI's Gene Expression Omnibus and is accessible through GEO Series accession number GSE162792 (https://www.ncbi.nlm.nih.gov/geo/query/acc.cgi?acc=GSE162792).
(XLSX)

**S2 Dataset. The metabolic gene expression profiles of WT and *alh-4(-)*.** The genes in following pathways were displayed: KEGG peroxisome pathway (sheet 1); mitochondrial fatty acid β-oxidation (sheet 2); peroxisomal fatty acid β-oxidation (sheet 3); tricarboxylic acid cycle (TCA) (sheet 4); lipid binding proteins for lipid transport; fatty acid desaturation (sheet 5).
(XLSX)

**S3 Dataset. The signal intensity of the lipids of interest identified by LC-MS in WT and *alh-4(-)*.** Lipids are grouped by the lipid class as indicated in the sheet name. The following information is provided: retention time (RT), the measured m/z value (m/z meas.), the measured neutral mass (M meas.), ions, name, molecular formula, annotation quality (AQ) defined by MetaboScape version 5.0 (Bruker Daltonics), annotation source, the relative mass deviation of the theoretical mass of the ion formula and the measured m/z value (Δm/z), collision cross section value (CCS) and the signal intensity detected in each replicate.
(XLSX)

**S4 Dataset. All numerical data that underlies figures with summary statistics (when applicable).** The name of the sheets is consistent with that of the figures.
(XLSX)

**S1 Text. Supplementary text on results pertaining to the determinants of organelle specific targeting of ALH-4 isoforms.**
(DOCX)

## Acknowledgments

We thank Meng Wang and King L. Chow for RNAi reagents and protocols. Jihong Bai, Christian Frøkjær-Jensen and Bob Goldstein for Mos1 and CRISPR reagents. Pui Shuen Wong at the HKUST Bioscience Central Research Facility for RNA sequencing and lipidomic analysis. Yan Li and Xingyu Yang for technical advice. Some strains were provided by the CGC, which is funded by NIH Office of Research Infrastructure Programs (P40 OD010440).

## Author Contributions

**Conceptualization:** Lidan Zeng, Ningyi Xu, Ho Yi Mak.

**Data curation:** Lidan Zeng, Xuesong Li, Ningyi Xu.

**Formal analysis:** Lidan Zeng, Xuesong Li, Christopher B. Preusch.

**Funding acquisition:** Ho Yi Mak.

**Methodology:** Lidan Zeng, Xuesong Li, Gary J. He.

**Supervision:** Tom H. Cheung, Jianan Qu, Ho Yi Mak.

**Writing – original draft:** Lidan Zeng, Ho Yi Mak.

**Writing – review & editing:** Lidan Zeng, Ho Yi Mak.

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
