## [Decision Letter · Decision Letter 0]

6 Apr 2021

Dear Dr Mak,

Thank you very much for submitting your Research Article entitled 'Nuclear receptors NHR-49 and NHR-79 promote peroxisome proliferation to compensate for aldehyde dehydrogenase deficiency in C. elegans' to PLOS Genetics.

The manuscript was fully evaluated at the editorial level and by independent peer reviewers. The reviewers appreciated the attention to an important topic but identified some concerns that we ask you address in a revised manuscript.

We therefore ask you to modify the manuscript according to the review recommendations. Your revisions should address the specific points made by each reviewer.

[LINK]

Yours sincerely,

Javier E. Irazoqui

Associate Editor

PLOS Genetics

Gregory P. Copenhaver

Editor-in-Chief

PLOS Genetics

Reviewer's Responses to Questions

**Comments to the Authors:**

Reviewer #1: In this manuscript, the authors report that nuclear receptors NHR-49 and NHR-79 play a role in regulating peroxisome proliferation to counteract aldehyde dehydrogenase deficiency. They first demonstrated in C. elegans that loss of aldehyde dehydrogenase alh-4 supresses lipid droplet (LD) expansion in daf-22 mutant worms. Next, they showed that ALH-4 associates with different organelles including peroxisomes through membrane anchoring. They also found a significant upregulation of peroxisome-related genes in alh-4(-) worms and that peroxisome proliferation was independent of ATGL-1 directed lipolysis. Next, they identified transcription factors NHR-49 and NHR-79 to be required in response to the lack of ALH-4 and that both regulate peroxisome proliferation. Their data suggest that intestinal ALH-4 regulates hypodermal peroxisome proliferation while NHR-49 and NHR-79 act cell-autonomously.

Overall, the experiments were well designed, executed, and included detailed analysis. The conclusions of the manuscript offer important perspectives on the role of ALH-4, NHR-49, and NHR-79 in peroxisome proliferation. However, there are few points that should be addressed to strengthen the authors’ manuscripts. I recommend the authors to perform additional experiments, to consider alternative explanations to rationalize their data, and/or to more/remove some of their experiments. I highly recommend the publication of the manuscript in PLOS genetics once my points are addressed by the authors.

Major points to address

(1) The forward genetic screen for suppressor mutants that reversed the daf-22 mutant phenotypes seems to be a high labor-intensive screen if phenotype was monitored by SRS and that the difference between WT and daf-22 is not that obvious. No details are provided in the method section. How many clones were screened?

(2) The data of LDs in alh-4(hj221) is not reported. The authors only show alh-4 RNAi in Fig. S1B. LDs in alh-4(hj221) must be reported as it is used in subsequent experiments where it is assumed that the mutant suppress daf-22 LDs phenotype.

(3) The authors reported triglyceride (TAG) levels in WT and alh-4(-) worms. I would have expected the authors to include daf-22 and daf-22; alh-4(-) mutants to their analysis.

(4) Pumping rate are abnormally high in Fig. S1F. It should be around 200-250 pumps/min in WT fed normal diet. Why are the rates double than most publications? Also, the authors should include more than 7 worms per condition.

(5) I find the section where the authors justify why GFP knockin strain retains normal ALH-4 function very confusing (lines 163-169). The authors argue using data that is described later in the text. I do not see the point to argue that their ALH-4 fusion protein is functional at this point by referring to data that is shown later in the text. This section would be more appropriate elsewhere either after Fig. 3 or in the discussion.

(6) The authors went into the distance to determine the subcellular localization of GFP::ALH-4 by using different organelle markers. I think this is very important and informative. However, I would like to see higher magnification to appreciate the localization of ALH-4 within cells.

(7) The number of worms should be clearly reported for all quantification based on fluorescence microscopy images. For instance in Fig. S4A,B, how many worms are reported per condition. The authors should distinguish technical from biological replicates in the figure as they are both reported with the same colour.

(8) It is not clear to me which genes are upregulated or if all of them are upregulated in Fig. 4B.

(9) The interaction seems rather very weak between NHR-49 and NHR-79 and the band corresponding to NHR-79 in the IP seems much higher in MW compared to input. Have the authors tried the reciprocal IP by IPing HA?

(10) Although reporting NHR-49 is required for reproduction of ALH-4 deficient worms is interesting. I do not see how it fits with previous results and how it adds to the understanding of the mechanism. In my view, it should be moved to supplementary data and certainly removed as the last sub-section of the result section.

Minor points to address

(1) I suggest the authors to change the title of line 113 to something more meaningful.

(2) Define LDs in line 115.

(3) Do the authors meant “sequencing” instead of “molecular cloning” in line 121?

(4) Fig. S1C should not appear after Fig. S1D-E. This is a recurrent issue through the manuscript and should be fixed. Figures and figure panels should appear in chronological order through the manuscript. Another example is Fig. S3A that is cited before Fig. S2. Fig. 3D,F are cited before Fig. 2B-F.

(5) The sentence “In agreement with our…” in lines 138-140 is repetitive as it is already reported above.

(6) The authors refer to alh-4 isoforms in lines 157-158 but Fig. 2A does not include any isoforms.

Reviewer #2: This manuscript by Dr. Ho Yi Mak and colleagues identifies a cross-tissue genetic pathway responsible for maintaining body fat mass in C. elegans. The pathway, involving intestinal ALH-4 and hypodermal peroxisomal proliferation downstream of NHR-49 and NHR-79 suggests that the latter responds to and is critical to the defense against excess levels of fatty aldehydes. It is particularly curious that the pathway would not be fully cell autonomous, that is excess aldehydes not dealt with in the tissues that they are generated in, and this adds to the interest of the work. The authors utilize state-of-the-art genetic and genomic methodologies that add strength to the conclusions put forward. In general, I find the conclusions sound, interesting, and robustly deduced. I do have modest concern about the interest level of the topic the broad readership of PLoS Genetics. Additionally, in some areas, the manuscript suffers from methodological and statistical inconsistencies that could be rectified to further strengthen conclusions:

Comment 1: It looks as though for fat mass, the relationship of alh-4 and daf-22 are parallel rather than epistatic; for lipid droplet alternatively the suppression looks epistatic. Please discuss in the text (Fig 1B&D).

Comment 2: Figure legend 1 says that two-way ANOVA with Tukey’s post hoc test was used, but then the figure legend also says t-tests were applied. Please clarify. Which is it? It is very important that multiple hypotheses be corrected for in the statistical analyses, particularly when multiple measurements are conducted as in 1B, D, E, and F.

Comment 3: what does the X axis represent in figs 2B-F (right panels)? distance? Percentage of total distance? Microns? Please clarify. Also, is it possible to show higher magnification images for co-localization? Is ALH-4 predicted to have an n-terminal mito targeting signal that the authors interfered with by knocking in the GFP at the n-terminus? What compartments didn’t GFP::ALH-4 associate with? The authors find evidence for all compartments having ALH-4 enrichment which could be interesting or, alternatively, confounded.

Comment 4: The TM domain deletion experiment (Fig. 3D-F) could have “failed” to rescue because of defective folding of the mutant protein rather than ineffective targeting to the proper cellular location (organelle or to membranes). This possibility should be addressed. Does the C- terminus harbor glycosylation sites that could be missing when deleted? These also could be involved in proper cellular targeting. The same could be true for the deltaGK mutant.

Comment 5: The relationship of ATGL to ALH-4 looks parallel rather than epistatic with regard to lipid droplet diameter in figure S4D-E), although the comparison between WT and alh-4 mutants on atgl-1 RNAi is not shown. Or if it is not significant, then the conclusion may be justified. As such the comparison is important to justify the conclusions.

Comment 6: Line 311: The loss of NHR-49 but not NHR-79”… should be “partially” corrected. The epistatic relationships with regard to PEX number and DAF-22 expression are much more convincing for epistasis whereas the LD data suggest the relationship is more complex than just nhr-49/79.

Comment 7: I am confused on the presentation of results in Figure 5. The authors indicate in the abstract that alh-4;nhr-49 double mutants are inviable, but proceed to analyze them in Figure 5B-G. The legend does not clearly indicate RNAi and the text suggests double mutants are analyzed. This must be clarified. Also unless the double alh-4;nhr-49 is inviable, it is not clear why this animal is not analyzed for the tissue-specific restorations of nhr-49. I do see in figure 8 that the double mutant makes very few progeny. It would have been interesting to see which of the tissue rescues in figure 5E/F also restore progeny production in the double mutant alh-4; nhr-49.

Minor issues:

I’m not sure if the conclusion is important given the genetic data, but if the upregulation of nhr-49 and nhr-79 at the mRNA level is significant and a point the authors want to substantiate, it should not be done with semiquantitative methodology like extraction of RNAseq data.

“number of progenies” should probably be “number of progeny”. In figure 8 and line 378, etc.

Reviewer #3: Mak and colleagues present a manuscript that describes a new genetic relationship between nhr-49 and nhr-79 in response to loss of the aldehyde dehydrogenase alh-4. This is a well designed and executed study that documents high quality data of the peroxisomal response to these genetic mutations. I found no issues with the methodology, appropriate statistics were used in most experiments, and the overall results are compelling. That said, I believe that there are three major concerns that should be addressed prior to acceptance (two of which can be done with simply editing of the manuscript).

Major concerns:

1. Overall meaning of the results. While the data is compelling, at the end of the manuscript the reader is left wondering why this is important in the grand scheme of things. Why does this genetic interaction exist? what is the purpose of engaging these NHRs in response to loss of a single ALH? Perhaps the manuscript would be aided by an enhanced summary figure that puts into context the physiological significance of the work?

2. Related to #1, the last figure identifies a physiological consequence of the genetic mutations study. The 30% reduction in progeny is a strong piece of data, but I am left wondering if this is a defect in oocyte, sperm, or overall change in reproductive tissue function. The author should test whether mating can rescue this phenotype and/or test if mutant male sperm have a defect in dominating fertilization.

3. Several of the claims in the manuscript are correlative when they could really be tested and shown to be causal.

3a. For example, the authors state "It is plausible that these catalases may protect alh-4(-) worms against exogenous hydrogen peroxide treatment (Fig S4B)." The authors should either test this directly in a ctl mutant background, or move statements like this to the discussion.

3b. Similarly the use of loss of function and gain of function alleles is a strength, but why was this specific nhr-49 gain of function allele chosen? Work from the Taubert lab suggests that different nhr-49gf alleles have specific activities. This should at a minimum be discussed further.

Minor concerns:

Without protein function (biochemical and enzymatic) the authors should limit their interpretations of the data to the genetic interactions.

The use of active and passive voice (and switching in and out of present/past tense) is awkward and should be edited.

**Have all data underlying the figures and results presented in the manuscript been provided?**

Reviewer #1: Yes

Reviewer #2: Yes

Reviewer #3: Yes

PLOS authors have the option to publish the peer review history of their article (what does this mean?). If published, this will include your full peer review and any attached files.

Reviewer #1: No

Reviewer #2: No

Reviewer #3: No

---

## [Decision Letter · Decision Letter 1]

2 Jun 2021

Dear Dr Mak,

We are pleased to inform you that your manuscript entitled "Nuclear receptors NHR-49 and NHR-79 promote peroxisome proliferation to compensate for aldehyde dehydrogenase deficiency in C. elegans" has been editorially accepted for publication in PLOS Genetics. Congratulations!

Please note that the reviewers had some final comments (see below) which you may want to consider as you prepare your final draft for the production team (the editorial team will not need to re-evaluate).

Yours sincerely,

Javier E. Irazoqui

Associate Editor

PLOS Genetics

Gregory P. Copenhaver

Editor-in-Chief

PLOS Genetics

Comments from the reviewers (if applicable):

Reviewer's Responses to Questions

**Comments to the Authors:**

Reviewer #1: The authors have addressed most critical concerns of the reviewers. The quality of the manuscript is significantly improved. I only have one remaining concerned that should be address.

In my initial comment #1, I raised that no details are provided in the method section regarding the screen of daf-22. The authors partially addressed it but I think that more details will be useful to fully appreciate their screen and to be able to conduct similar screens. First, C1-BODIPY-C12 doesn't stain LD specifically. C1-BODIPY-C12 is usually used as marker of fatty acid trafficking. Why was it used instead of BODIPY 493-503? I am still unsure how the authors managed to screen for a small difference in LDs using a dissecting fluorescence microscope. What was the threshold used to define a mutant as suppressors? The number of screened wormed should be indicated in the method as well as the number of worms selected in each steps of the screen.

Reviewer #2: The authors have suitably addressed all of my concerns in this revision. I do not completely agree with the response to my comment on correction for multiple hypothesis testing, particularly in figure 1F, but the authors justify their approach appropriately. As long as the methods are fully disclosed in the legend of the figure this is acceptable. Typically the P value is adjusted for the number of measurements made or the numbers of hypotheses tested, e.g. in RNAseq and in LC-MS or GC-MS for metabolites.

Reviewer #3: the authors have addressed the previous concerns raised.

**Have all data underlying the figures and results presented in the manuscript been provided?**

Reviewer #1: Yes

Reviewer #2: Yes

Reviewer #3: Yes

PLOS authors have the option to publish the peer review history of their article (what does this mean?). If published, this will include your full peer review and any attached files.

Reviewer #1: No

Reviewer #2: No

Reviewer #3: No

**Data Deposition**

http://datadryad.org/submit?journalID=pgenetics&manu=PGENETICS-D-21-00308R1

**Press Queries**

---

## [Editor Report · Acceptance letter]

1 Jul 2021

PGENETICS-D-21-00308R1 

Nuclear receptors NHR-49 and NHR-79 promote peroxisome proliferation to compensate for aldehyde dehydrogenase deficiency in C. elegans 

Dear Dr Mak, 

We are pleased to inform you that your manuscript entitled "Nuclear receptors NHR-49 and NHR-79 promote peroxisome proliferation to compensate for aldehyde dehydrogenase deficiency in C. elegans" has been formally accepted for publication in PLOS Genetics! Your manuscript is now with our production department and you will be notified of the publication date in due course.

With kind regards,

Andrea Szabo

PLOS Genetics

On behalf of:
